# On Next-Token Prediction in LLMs: How End Goals Determine the Consistency of Decoding Algorithms

## Abstract

Probabilistic next-token prediction trained using cross-entropy loss is the basis of most large language models. Given a sequence of previous values, next-token prediction assigns a probability to each possible next value in the vocabulary. From this, there are many ways to turn these next-token predictions into token sequences. This paper examines a few of these algorithms (greedy/lookahead decoding, random sampling, and temperature-scaled random sampling) and studies their consistency with respect to the user end goals of information retrieval and creative generation through encoding these goals as loss functions. Although the consistency of surrogate losses with respect to a target loss function is a well researched topic, we are the first to study it in the context of LLMs (to the best of our knowledge). We find that, so long as next-token prediction converges to its true probability distribution, random sampling is consistent with outputting sequences that mimic sampling from the true probability distribution. For the other goals, such as minimizing the 0-1 loss on the entire sequence, we show that deterministic decoders have the edge over stochastic decoders. From these results, we see that there is a dichotomy created between the goals of information retrieval and creative generation for the decoding algorithms. This shows that choosing the correct decoding algorithm based on the desired goal is extremely important and many of the ones used are lacking theoretical grounding in numerous scenarios. While there has been evidence for this empirically, this paper gives rigorous theoretical grounding to these results.

## 1 INTRODUCTION

The basis for nearly all large language models today is next-token prediction trained by minimizing the cross entropy loss function (Radford et al., 2018; Brown et al., 2020; Chowdhery et al., 2023; Touvron et al., 2023; Devlin et al., 2019). However, next-token prediction only gives probabilities for the next token. Many tasks, such as machine translation or text generation, require an output of a token *sequence*. Thus, we must have some decoding algorithm that takes these next-token predictions and outputs a sequence. With a large amount of test-time computation being used in state-of-the-art models (Jaech et al., 2024; Guo et al., 2025), the mathematical foundations of these algorithms are of great interest.

In this paper, we analyze the behavior of next-token prediction and how the choice of decoding algorithms can impact the asymptotic optimality of the outputs. This work can be thought of as studying *surrogate loss consistency*: we minimize a surrogate loss function (cross entropy on next-token prediction), but are interested in a different target loss function (e.g., Hamming loss between predicted and correct sequence). This notion of consistency has been extensively studied in machine learning. Bartlett et al. (2006) showed results for binary classification where they minimize a surrogate loss function and show consistency with respect to the 0-1 loss. Tewari & Bartlett (2007) extend this approach to multi-class classification. Gao & Zhou (2011); Koyejo et al. (2015); Wu & Zhu (2020) all have worked on consistency for multi-label classification.

There has also been research into next-token prediction and decoding algorithms. Saunshi et al. (2021) investigates how linearly transforming next word prediction can predict text classification. Li et al. (2024) studies what can be learned by a single attention layer for next-token prediction. Snell et al. (2024); Wiher et al. (2022); Shi et al. (2024) all investigate a few types of decoding algorithms and empirically evaluate

them. There has also been much research on how next-token prediction learns (Bachmann & Nagarajan, 2024; Lin et al., 2025; Thrampoulidis, 2024), but, as far as we are aware, we are the first to investigate the consistency of next-token prediction and these decoding algorithms.

It is standard to analyze consistency in an asymptotic setting of sufficiently large sample sizes and models where the surrogate loss has been fully minimized. We therefore assume that our next-token predictor converges to the true next-token distribution and we only have query access to it. Note that this emulates the training of our next-token predictor as asymptotic minimization of cross-entropy results in correct next-token distributions. Given this, we investigate our decoding algorithms with regards to two high-level goals central to how large language models are used today: *information retrieval*, where the user is looking for a "correct answer" and *creative generation*, where the user is looking for new samples from the distribution of human language (Paaß & Giesselbach, 2023; Petroni et al., 2019; Brown et al., 2020). We do this by minimizing a loss function that acts as a proxy for these goals: the N-gram Hamming loss for correct information retrieval and the cross entropy loss for the entire sequence for generating samples.

This paper gives a framework to study the consistency of various decoding algorithms with respect to different high level goals. We show that only *deterministic* decoding algorithms can be consistent for the N-gram Hamming loss, however, these have infinite loss for the cross entropy loss for any non-deterministic true sequence output distribution. Therefore, *stochastic* decoders are necessary to have a non-infinite loss for the cross entropy objective, but fail at consistency for the N-gram Hamming loss. This shows there is no one-size-fits-all decoding algorithm and one must adapt their decoder to their desired goal. While this dichotomy has been empirically studied (Wiher et al., 2022; Shi et al., 2024), we are the first to give theoretical justification to this phenomenon as far as we are aware. There has also been a recent line of work on adaptive decoding strategies (Zhu et al., 2024b;a; Dhuliawala et al., 2024). These decoding strategies change the decoder output distribution to be more or less determinsitic-like depending on some criteria which aligns itself with the information retrieval versus creative generation dichotomy. Therefore, our theoretical results are consistent with these recent empirical findings.

We also show that there is no consistent polynomial-time decoding algorithm for all output distributions for the N-gram Hamming loss and we create a small Markov chain experiment to see the optimality of deterministic decoders. For the cross entropy loss, we show random sampling is consistent for all probability distributions over the sequence outputs. Finally, we give a rate for the suboptimality gap of the expected risk for temperature scaling with respect to the temperature parameter.

This paper is organized as follows: Section 2 goes over the requisite background and notation needed for this paper, including the decoding algorithms studied in this paper. Section 3 discusses the problem set up. Section 4 goes over the case where the goal is information retrieval. Finally, section 5 shows results for when the goal is creative generation.

## 2 Notation and background

For notation, we will use $\mathcal{Y}$ as an output space, $\mathcal{X}$ as an input space, and $\ell$ as a loss function. Since the outputs are sequences, $y \in \mathcal{Y}$ will refer to the entire sequence, $y_i$ will refer to index $i$ in the sequence, and $y_{[i]}$ will refer to the subsequence from indices 1 to $i$. In general, $[j]$ is the ordered set $(1, 2, \ldots, j)$ and $y_{i:j} = y_i y_{i+1} \ldots y_j$. For two strings, the $+$ operator will mean concatenation. For probabilities, when one sees $p(v \mid y_{[i-1]})$, this is the conditional probability of the $v$ at index $i$ given the sequence $y_{[i-1]}$. We will often write this as $p(y_i \mid y_{[i-1]})$. To keep consistent with this notation, we will write $p(y_i)$ as the marginal of the distribution at index $i$ for the token $y$.

### 2.1 Expected risk

Given a loss function $\ell$, a probability distribution $p$ over inputs $\mathcal{X}$ and outputs $\mathcal{Y}$, and a hypothesis $h : \mathcal{X} \to \mathcal{Y}$, the expected risk is normally defined as follows:

$$R(h, p, \ell) = \mathop{\mathbb{E}}_{(x,y) \sim p} [\ell(h(x), y)].$$

This value is oftentimes what is trying to be minimized when training a machine learning algorithm (Shalev-Shwartz & Ben-David, 2014; Bartlett et al., 2006; Tewari & Bartlett, 2007; Gao & Zhou, 2011), however, this set up does not have the required granularity needed for our purposes. There has been work that has modified the expected risk so that it can fit their use cases, such as needing a "pred" function to convert the output of their algorithm into a prediction (Tewari & Bartlett, 2007; Ramaswamy et al., 2013; Ramaswamy & Agarwal, 2016). We will then also reformulate the expected risk so that it fits to our problem.

We assume we have access to a next-token predictor, which can only output conditional probabilities for the next token given the previous tokens and input. The notation for this will be $p_{ntp}(y_i \mid y_{[i-1]})$ for every $y_i$ in our vocabulary. These conditional distributions naturally induce a unique probability distribution on the entire sequence, thus we will interchangeably refer to $p_{ntp}$ as probability distribution itself. A $ntp$ subscript on a probability distribution is used to emphasize that the distribution is being used as a next-token predictor. Given $p_{ntp}$, we require a decoding algorithm $\mathcal{D}$, which takes in an input $x \in \mathcal{X}$ and a next-token predictor $p_{ntp}$ ($\mathcal{D}$ only has access to conditional distributions of $p_{ntp}$) and uses them to output a sequence $\hat{y} \in \mathcal{Y}$. Since $\mathcal{D}$ can have internal randomness, we will refer to the distribution of outputs produced by $\mathcal{D}$ as $p_{\mathcal{D}(p_{ntp})|x}$. We will then define our expected risk as follows:

$$R(\mathcal{D}, p, p_{ntp}, \ell) = \mathop{\mathbb{E}}_{x \sim p_x} \left[ \mathop{\mathbb{E}}_{y \sim p|x, \hat{y} \sim p_{\mathcal{D}(p_{ntp})|x}} [\ell(\hat{y}, y)] \right],$$

where $p \mid x$ is the conditional distribution of the output $y$ given an input $x$.

## 2.2 Decoders

We will look at 3 types of decoding algorithms: $K_T$-lookahead decoding, random sampling, and temperature-scaled random sampling.

### 2.2.1 $K_T$-lookahead

The word "lookahead" has been used in a few different ways in LLM decoding (Snell et al., 2024; Fu et al., 2024). Here, we will define the $K_T$-lookahead algorithm as a generalization of the well-known "greedy" decoding algorithm (Shi et al., 2024; Wiher et al., 2022). For choosing the next token(s), we will find all $K$-length combinations of our tokens and then keep the first $T$ tokens of the maximum $K$-length sequence.

---

**Algorithm 1** $K_T$-lookahead

---

**Require:** $L \in \mathbb{N}$, $K \leq L$, $T \leq K$, $p_{ntp}(\cdot \mid \cdot)$, Vocabulary $\mathcal{V}$

$\quad y \leftarrow$ "''

$\quad$ **while** length$(y) < L$ **do**

$\quad\quad c \leftarrow \max\{K, L - \text{length}(y)\}$

$\quad\quad y' = \arg\max_{v_1,\ldots,v_c \in \mathcal{V}}\{p_{ntp}(v_1 \mid y)p_{ntp}(v_2 \mid y + v_1)\ldots p_{ntp}(v_c \mid y + v_{1:c-1})\}$

$\quad\quad H \leftarrow \min\{T, L - \text{length}(y)\}$

$\quad\quad y \leftarrow y + y'_{[H]}$

$\quad$ **end while**

---

We note that $K = T = 1$ is greedy decoding.

### 2.2.2 Random sampling

Since next-token prediction outputs probabilities, random sampling will choose the next token given these conditional probabilities.

---

**Algorithm 2** Random Sampling

---

**Require:** $L \in \mathbb{N}$, $p_{ntp}(\cdot \mid \cdot)$, Vocabulary $\mathcal{V}$
  $y \leftarrow$ "''
  **while** length$(y) < L$ **do**
    $y_{new} \sim p_{ntp}(\cdot \mid y)$
    $y \leftarrow y + y_{new}$
  **end while**

---

### 2.2.3 Temperature-scaled random sampling

Temperature scaling is where one scales the next-token probabilities to encourage or discourage exploration of the space. It is used in almost all, if not all, large language models (Brown et al., 2020; Achiam et al., 2023; Chowdhery et al., 2023; Touvron et al., 2023).

Normally the probabilities are found by using a softmax on logits $z_i$. Temperature scaling is then done by using softmax on $z_i/T$, where $T$ is the temperature parameter. However, using only the probabilities themselves, we can do temperature scaling using temperature $\gamma$ as:

$$p_{scaled}(y_i \mid y_{[i-1]}, x) = \frac{p(y_i \mid y, x)^\gamma}{\sum_{v \in \mathcal{V}} p(v \mid y, x)^\gamma}.$$

We show the equivalence in Appendix B.1.

For this decoding algorithm, we will be randomly sampling the next token from the temperature-scaled distribution.

---

**Algorithm 3** Temperature Scaled Random Sampling

---

**Require:** $L \in \mathbb{N}$, $\gamma > 0$, $p_{ntp}(\cdot \mid \cdot)$, Vocabulary $\mathcal{V}$
  $y \leftarrow$ "''
  **while** length$(y) < L$ **do**
    $p_\gamma(v \mid y) = \frac{p_{ntp}(v|y)^\gamma}{\sum_{u \in \mathcal{V}} p_{ntp}(u|y)^\gamma}$
    $y_{new} \sim p_\gamma(\cdot \mid y)$
    $y \leftarrow y + y_{new}$
  **end while**

---

## 3 Problem setup

Let $\mathcal{X}$ be a general input space. Let $\mathcal{V}$ be a vocabulary, $*$ be a null character, and let $L \in \mathbb{N}$. Then, let $\mathcal{Y} \subseteq \{y_1 y_2 \ldots y_j \underbrace{* * *}_{L-j \text{ indices}} \mid y_i \in \mathcal{V}, j \leq L\}$ be our sequence output space. Each $y \in \mathcal{Y}$ is thus a sequence padded to a finite maximum length using the null character. We will also assume for any next-token predictor, if the current string has $*$ in it, all mass for the next token is at $*$. This is done as $*$ represents empty space and is only used in the analysis to simplify dealing with strings of different lengths. The Transformer architecture has a maximum sequence length it can output, thus this set up does not lose any generality to modern day large language models.

Let us represent the true probability distribution as $p^*$ over $\mathcal{X} \times \mathcal{Y}$. Given an initial next-token predictor $p_{ntp}^0$, we will assume that it is iteratively updated using cross entropy on the next-token distributions. Let each new iteration be $p_{ntp}^i$.

**Assumption 3.1.** $\forall y \in \mathcal{Y}$, $\forall i \in [L]$    $p_{ntp}^i(y_i \mid y_{[i-1]}) \to p^*(y_i \mid y_{[i-1]})$ *in KL-Divergence.*

It is standard to study surrogate loss consistency when the surrogate loss is asymptotically minimized (Gao & Zhou, 2011; Tewari & Bartlett, 2007). It can be easily seen that minimizing cross entropy implies the

KL-Divergence is 0. From a practical standpoint, this assumption is credible as, given proper data, modelling, and updating, the next-token conditional distributions will converge to the true conditional distribution through the minimization of cross entropy. We then show in Appendix B.2 that this also implies $p_{ntp} \to p^*$ in KL-Divergence as well.

Now, given $p_{ntp}^i \to p^*$, this paper studies when our decoding algorithms have the property

$$R(\mathcal{D}, p^*, p_{ntp}^i, \ell) \to \inf_{h:\mathcal{X}\to\mathcal{Y}} R(h, p^*, \ell).$$

## 4 Consistency for N-gram Hamming loss

Historically, N-grams have been important in sequence metrics like the BLEU score (Papineni et al., 2002) and the ROUGE-N score (Lin, 2004). N-grams are used to segment a sequence into portions evaluate the correctness of each portion. We will take this idea and define a new loss function, which we call the N-gram Hamming loss. Mathematically, we define it as:

$$\sum_{i=1}^{L-N+1} \mathbf{1}_{\{\hat{y}_{i:i+N-1} \neq y_{i:i+N-1}\}}.$$

For $N = 1$ this is the Hamming loss and for when $N = L$ we have the 0-1 loss, which themselves are two canonical loss functions in machine learning. Intermediate losses when $N \in [2, L-1]$ might be useful in their own right, but here we consider them as a mathematically tractable representative for the various N-gram based metrics used in sequence learning.

We want to determine for which probability distributions will our decoding algorithms always produce the optimal output for all sets of inputs with positive measure. Below we show what is optimal for the N-gram Hamming loss:

**Lemma 4.1.** *Let $p$ be a probability distribution over output sequences and let*

$$g(y) = \sum_{i=1}^{L-N+1} p(y_{i:i+N-1}).$$

*Then, the optimal output for N-gram Hamming is*

$$\arg\max_y \{g(y)\}.$$

The proof of is left to the Appendix B.4 for ease of presentation. Note how this generalizes the already known optimal outputs for the Hamming and 0-1 loss (Dembczyski et al., 2010).

### 4.1 Exchanging consistency For optimality

Here we give a useful result that will allow us exchange the limit and expectation in the expected risk, given a decoder meets the assumptions needed.

**Proposition 1.** *Suppose $p_{\mathcal{D}(p_{ntp})|x}$ is the probability distribution of the output of $\mathcal{D}(p_{ntp}) \mid x$. Then, given an $M$-bounded loss function $\ell$ and*

$$\forall x \in \mathcal{X}, \forall y \in \mathcal{Y} \lim_{i\to\infty} \left( p_{\mathcal{D}(p_{ntp}^i)|x}(y) - p_{\mathcal{D}(p_{ntp}^*)|x}(y) \right) = 0.$$

*Then*

$$\lim_{i\to\infty} R(\mathcal{D}, p^*, p_{ntp}^i, \ell) = R(\mathcal{D}, p^*, p_{ntp}^*, \ell).$$

We leave the proof to Appendix B.3 for ease of presentation. This allows us to deal with the *optimality* of our decoding algorithms given a true sequence distribution $p^*$ instead of the *consistency* of a sequence $p^i$. The assumption is also very reasonable, if not even a desirable trait for a decoder to have: it says that as $p^i$ converges to $p^*$, the probability of our decoder outputting any sequence using $p_{ntp}^i$ should converge to the probability of our decoder outputting that sequence under $p_{ntp}^*$.

### 4.2 Optimal decoding for all probability distributions is not in polynomial time

To motivate the usage of various decoding algorithms for the N-gram Hamming loss, we will show that, even if we have access to next-token predictons from the true sequence distribution ($p^*$), there does not exist a polynomial-time (in sequence length $L$) decoding algorithm that is optimal for all probability distributions. We show in Section 4.4 that stochastic decoders (i.e., decoders that can sometimes choose one value or another depending on internal randomness) can be not optimal so long as they put non-zero mass on an non-optimal output. Thus we are left with two types of decoders: deterministic decoders and stochastic decoders that put all the probability mass on optimal outputs. Let us call the latter optimal stochastic decoders.

**Theorem 4.2.** *Let $\mathcal{V}$ be a vocabulary and let $\mathcal{Y}$ have maximum length $L$. Let $p$ be such that*

$$\forall y \in \mathcal{Y}, \ \forall i \in L \quad p(y_i|y_{[i-1]}) = \frac{1}{|\mathcal{V}|}.$$

*Then, any optimal deterministic or optimal stochastic decoder algorithm $\mathcal{D}$ for the N-gram Hamming loss must have a runtime of at least $C(|\mathcal{V}|^L - 1)$, assuming queries to the next-token predictor take $C$ time.*

The proof is left to Appendix B.5. The idea of the proof relies on the pigeonhole principle. If there are an exponential amount of values $p(v|y_{[j-1]})$ that can be queried, if we only query a polynomial amount of them, there are many adversarial probability distributions that fit the queried values, but have different optimal values. Even though the proof is done on the uniform distribution, one can see how this idea can fit a large class of sequence probability distributions.

**Corollary 4.2.1.** *Optimal decoding of the N-gram Hamming problem takes exponential time in $L$ assuming black-box access to next-token probabilities.*

*Proof.* Since accessing from memory is assumed to be constant time, we have shown there is a distribution that will take $\Omega(|\mathcal{V}|^L)$ runtime. This, combined with the results in Section 4.4, gives us our result. $\square$

In modern large language models, these next-token probabilities are done using calls to a Transformer architecture, which do not have $O(1)$ runtime. However, for the N-gram Hamming loss, we see that next-token prediction has an exponential lower bound for optimality that no amount of Transformer optimization can fix.

### 4.3 $K_T$-lookahead decoding

For this subsection, ties are a curse. When choosing the next token(s), if we have at least 2 sequences in our next token(s) arg max such that at least 1 puts the algorithm not on a path to an overall sequence arg max, then there is no way for our algorithm to be optimal for all probability distributions. This is because there are two ways to break ties: deterministic breaks or random breaks. For a deterministic tiebreak, we can adversarially create a probability distribution where we choose wrong. For random tiebreaks, we show in in Section 4.4 that it also can not be optimal. Therefore, we will only look at the class of probability distributions where we will not run into any ties in any of our arg maxs. Let this class be called $\mathcal{P}$.

We note that restricting to this set means that we can not use the example given in the proof of optimal decoding requiring exponential time. However, the proof does not rely on ties, it instead relies on having no few conditional probabilities $p(v|y_{[i-1]})$ being large enough to make all the others at that level irrelevant. Thus, one can imagine extremely small perturbations to the example's conditional distributions such that the distribution is in $\mathcal{P}$, but is still very close to the uniform distribution. The rest of the proof would then work out the same.

**Lemma 4.3.** *$K_T$-lookahead decoding meets the criteria to use Proposition 1*

The proof is left to Appendix B.6 for ease of presentation. Using this lemma, we only need to concern ourselves with $p^*$ when trying to show consistency.

Now, in order for $K_T$-lookahead decoding to be optimal for the N-gram Hamming loss, we give a reframing of $K_T$-lookahead in terms of what probability distributions it is optimal for below:

**Proposition 2.** *Let us have a probability distribution $p^* \in \mathcal{P}$ over $\mathcal{X} \times \mathcal{Y}$ and let*

$$C = \{x \mid \arg\max_y \sum_{i=1}^{L-N+1} p^*(y_{i:i+N-1} \mid x) = y^\dagger \; where$$

$$y^\dagger_{(Tc+1):\min\{(Tc+T),L\}} =$$

$$\left( \arg\max_{y_{(Tc+1):\min\{(Tc+K),L\}}} \{p^*(y_{(Tc+1):\min\{(Tc+K),L\}} \mid x, y^\dagger_{[Tc]})\} \right)_{[T]}$$

$$for \; c \in \mathbb{Z}_+, Tc \leq L - T\}.$$

*Then, $K_T$-lookahead is N-gram Hamming loss optimal for $p^*$ iff*

$$p_x^*(C) = 1.$$

The proof is left to Appendix B.7.

Since $K_T$-lookahead is a greedy algorithm, this characterization can be interpreted as it only being optimal for probability distributions that lend themselves well to greedy algorithms. Thus, no matter how perfect the next-token predictor is, we are simply running a greedy algorithm and these algorithms will fall into the same traps greedy algorithms have been known to fall into. For example, in Appendix B.8 we give a Markov chain that is not optimal by exploiting the characterization above.

Given this, it is natural to then ask how often do these decoders run into such a problem. We set up a simulation study on Markov chains to empirically test optimality. We create each graph by having its starting distribution and each transition distribution be Dirichlet distributed with the parameters all being the same value, $\alpha$. We group each graph by the $\alpha$ parameter used and then take the average amount of times $K_T$-lookahead was optimal. Each group has 200 graphs. We do note, while improbable, there can be ties in the $K_T$-lookahead arg max in our simulations and the ties are broken by which path was seen first. More details on the simulation study can be found in Appendix A.1.

While this is a simple experiment, we note that our results show that we can not do much more. Since we are looking for optmality and there is no polynomial-time optimal decoder, we must use an exponential-time algorithm to find the optimal sequence. This immediately narrows the experiments we can run since we can only use a model with a restrictively small vocabulary and maximum sequence length.

In Figure 1, we plot the percent of times $K_1$-lookahead was optimal in a group for the 1-gram Hamming loss against the average KL-Divergence from the uniform distribution for that group. We can see that it does not do well no matter how short the sequence is. Even for short sequences with low entropy distributions (very "peaky" distributions), $K_1$-lookahead decoding still guesses wrong about 10% of the time.

The next figure, Figure 2, we see how $K_1$-lookahead does for the $L$-gram Hamming loss. When $K = L$, $K_1$-lookahead will be optimal, which is why each of them has one line that is perfect. For the rest of the lines, we see they do better than they did for the Hamming loss, which is interesting in its own right. This trend is generally seen when looking at the other values of $N, L, K$ as well. We leave a more thorough explanation and analysis of our simulations to Appendix A.1, where we also empirically analyze $K_K$-lookahead decoding as well.

A beam search is, instead of greedily choosing at each step in the lookahead algorithm, one keeps the top $B$ sequences at each step and uses them for the next steps. Once at the end, then the beam search chooses the best out of the B outputs (Shi et al., 2024; Wiher et al., 2022). We can easily extend our reframing of $K_T$-lookahead to a $K_T$-lookahead beam search. Let arg max B be the set of the top $B$ values.

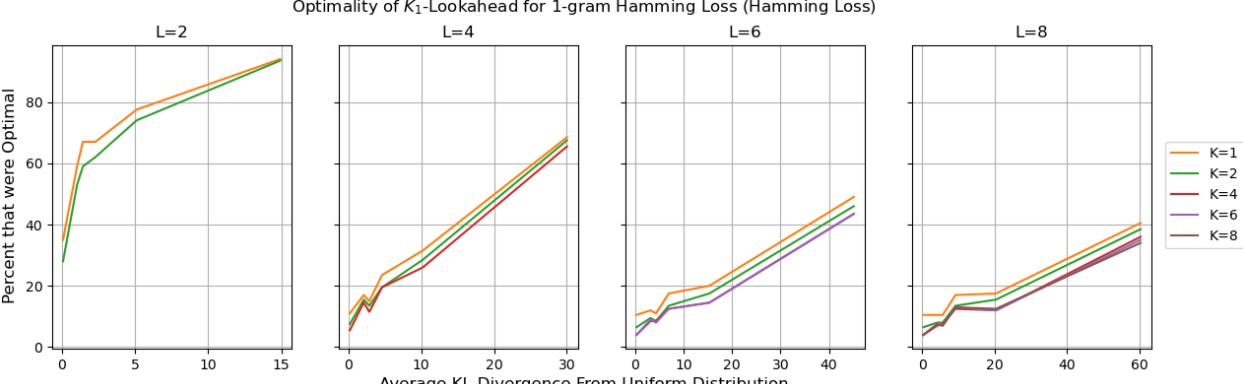

Figure 1: A plot of the amount of trials $K_1$-lookahead was optimal for the 1-gram Hamming loss (the Hamming loss). Each point represents the average optimality over 200 randomly generated Markov chains with a set amount of nodes and Dirichlet parameter $\alpha$. Smaller $\alpha$s create more "peaky" distributions and thus have higher KL-divergence from the uniform distribution, while larger $\alpha$s create more uniform distributions. There were 8 nodes in each Markov chain for this figure and the sequence length goes up by two as one moves right in the plots.

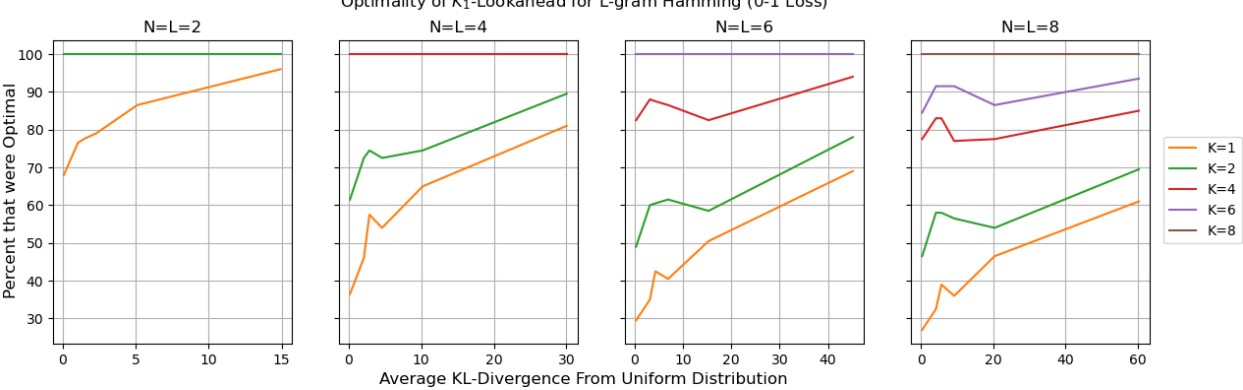

Figure 2: A plot of the amount of trials $K_1$-lookahead was optimal for the $L$-gram Hamming loss (the $0-1$ loss). The same setup as Figure 1 otherwise.

**Corollary 4.3.1.** *Let us have our decoder be a $K_T$-lookahead beam search with beam width $B$. Let us have a probability distribution $p^* \in \mathcal{P}$ over $\mathcal{X} \times \mathcal{Y}$ and let*

$$C = \{x \mid \arg\max_y \sum_{i=1}^{L-N+1} p^*(y_{i:i+N-1} \mid x) = y^\dagger \text{ where}$$

$$y^\dagger_{(Tc+1):\min\{(Tc+T),L\}} \in$$

$$\left( \arg\max_{y_{(Tc+1):\min\{(Tc+K),L\}}} B \quad \{p^*(y_{(Tc+1):\min\{(Tc+K),L\}} \mid x, y^\dagger_{[Tc]})\} \right)_{[T]}$$

$$\text{for } c \in \mathbb{Z}_+, Tc \leq L - T\}.$$

*Then, $K_T$-lookahead is N-Hamming loss optimal for $p^*$ iff*

$$p^*_x(C) = 1.$$

*Proof.* The same as Proposition 2, but with $\arg\max$ B instead of $\arg\max$. □

Going back to greedy $K_T$-lookahead, let $\mathcal{P}_{K,T,N}$ be all probability distributions where $K_T$-lookahead is optimal with respect to N-gram Hamming.

One could expect that increasing $K$ would monotonically increase $\mathcal{P}_{K,T,N}$ since the decoder is "looking" farther into the future. One would also expect that decreasing $T$ would monotonically increase $\mathcal{P}_{K,T,N}$ since taking less tokens now will allow us to reconsider later when we have more information. However, we show below that neither of these are generally the case.

**Proposition 3.** *Let $K_1, K_2 \in [L-1]$, where $K_1 < K_2$. Then $\forall T_1 \in [K_1]$, $\forall T_2 \in [K_2]$, and $\forall N \in [L]$, we have $\mathcal{P}_{K_1,T_1,N} \not\subset \mathcal{P}_{K_2,T_2N}$.*

**Proposition 4.** *Let $K \in \{2, 3\ldots, L-1\}$, $N \in [L]$ and $T \in [K-1]$. Then, if $K < L-T$, we have $\mathcal{P}_{K,T+1,N} \not\subset \mathcal{P}_{K,T,N}$*

These proofs are left to Appendix B.9 and B.10, respectively. For Proposition 4, when $K \geq L-T$, there can exist monotonicity. For example, in Appendix B.10 we show it for when $N = L$.

These propositions show that the $K$ and $T$ hyperparameters have no general monotonicity and their optimality is distribution specific. In the case of large language models, since conditioning on different prompts changes the distribution, this shows that the optimal hyperparameters of $K_T$-lookahead are input dependent.

### 4.4 Stochastic decoders

Below we show any decoder that is stochastic in any way is not optimal, so long as there is a non-zero probability of choosing a sequence that is not optimal. The proof is left to Appendix B.11.

**Proposition 5.** *Let our loss function be the N-gram Hamming loss. Let $p$ be a probability distribution over $\mathcal{X} \times \mathcal{Y}$. Then, any stochastic decoder that has a non-zero probability of outputting a $\hat{y} \in \mathcal{Y}$ where $\hat{y} \notin \arg\max_y g(y)$ for a set of inputs that have non-zero probability is not optimal.*

#### 4.4.1 Random sampling and temperature-scaled random sampling

We show in Appendix B.12 that both of these decoding algorithms meet the criteria for Proposition 1. Thus, since each of the runtimes of these scale linearly with $L$, by Proposition 1 and Theorem 4.2, we have that neither of these decoders are consistent for all probability distributions. In fact, so long as $\gamma \neq \infty$, neither of these are consistent for any non-uniform or non-deterministic probability distribution in $\mathcal{P}$ by Proposition 5.

## 5 Consistency for sample generation

One valid goal of a large language model is to sample responses from the distribution of human speech Paaß & Giesselbach (2023). We know that for any two probability distributions, minimizing cross entropy implies they will be equal to each other. Therefore, if we want our goal to be sampling from a true sequence probability distribution, the cross entropy loss on the entire sequence is a natural choice. In this section we will not only use cross entropy to train our next-token predictor, but also use it as a loss function for our sequential output.

Given a decoding algorithm $\mathcal{D}$ and input $x \in \mathcal{X}$, the cross entropy loss is defined as follows:

$$CE(p^* \mid x, \ p_{\mathcal{D}(p_{ntp})|x}) = \mathop{\mathbb{E}}_{y \sim p^*|x} \left[ -\log\left(p_{\mathcal{D}(p_{ntp})|x}(y)\right) \right].$$

### 5.1 Deterministic decoders

For deterministic decoding algorithms, it is easy to see by the definition of cross entropy that for any non-deterministic probability distribution, these decoding algorithms will have infinite cross entropy. Thus, they are not consistent.

## 5.2 Random sampling

In the N-gram Hamming setting, we saw that random sampling is not consistent for all non-deterministic or non-uniform probability distributions. Here, however, we will show it is consistent for every probability distribution. The proof is left to Appendix B.13.

**Proposition 6.** *Random sampling is always consistent under the cross entropy loss function in our setting.*

We note that, once we set up the surrogate loss framework, this essentially follows from the definition of autoregressive modeling. We believe this shows how natural this framework fits within this setting.

## 5.3 Temperature-scaled random sampling

From the previous section we can see that when $\gamma = 1$, we know temperature-scaled random sampling is consistent with any true probability distribution. However, below we show it is not consistent for nearly all true probability distributions when $\gamma \neq 1$.

**Proposition 7.** *When $\gamma \neq 1$, temperature-scaled random sampling is only optimal for uniform or deterministic distributions.*

The proof is left to Appendix B.14.

We next show how the expected risk increases as $\gamma$ changes. For convenience, we assume below that for $\mathcal{Y} = \mathcal{V}^L$. The proof method works in generality, however, the resulting $\log(|V|)$ term in the lower bound would be much less interpretable. A more in-depth explanation is given at the end of Appendix B.15.

**Proposition 8.** *Let $p$ be a probability distribution over $\mathcal{Y}$ and let $p^\gamma$ be our temperature-scaled random sampled distribution with respect to $p$. Let opt be the minimum expected cross entropy obtainable with respect to $p$. Then, there exists constants $C_1, C_2, C_3 \in \mathbb{Z}_+$ that depend only on $p$ such that we have:*

*For $\gamma > 1$:*

$$\gamma C_1 - opt \leq \mathop{\mathbb{E}}_{y \sim p} \left[ -\log\left(p^\gamma(y)\right) \right] - opt \leq$$

$$\gamma C_3 + L \log\left(|V|\right) - opt.$$

*For $\gamma < 1$:*

$$\left(L \log\left(|V|\right) - opt\right) - \gamma C_2 \leq \mathop{\mathbb{E}}_{y \sim p} \left[ -\log\left(p^\gamma(y)\right) \right] - opt \leq$$

$$\left(L \log\left(|V|\right) - opt\right) + \gamma C_3.$$

We leave the proof to Appendix B.15.

We require two bounds due to how the behavior changes from when $\gamma < 1$ to $\gamma > 1$. We know that as $\gamma$ approaches $\infty$, our distribution becomes closer and closer to a point mass, thus our cross entropy goes to infinity. The inequalities set forth in when $\gamma > 1$ shows our loss goes to infinity at a rate of $\gamma$.

As $\gamma$ approaches 0, we know our distribution will get closer and closer to the uniform distribution. Thus, our cross entropy should go to $-\log\left(|V|^{-L}\right) = L \log\left(|V|\right)$, which we can see through the inequalities when $\gamma < 1$ that it also goes to $L \log\left(|V|\right)$ as well at a rate of $\gamma$.

From this, we see temperature scaling behaves asymptotically as is expected in its scaling parameter and it does so linearly.

## 5.4 Proper losses

A proper loss function with regards to this work is a loss function whose minimum is obtained by the true distribution. One well-used proper loss is the cross entropy loss and others include the Brier loss and the spherical loss (Vovk, 2015).

**Corollary 5.0.1.** *All the results regarding cross entropy loss from Sections 5.1 and 5.2 hold for any proper loss.*

*Proof.* Cross entropy is a proper loss and since random sampling decoding is consistent with it, we have that random sampling is consistent with respect to any proper loss. Further, since we see that no deterministic function is consistent for non-deterministic probability distributions, we have that they do not limit to the true distribution. Thus, they can not be consistent for other proper losses as well. □

This also shows how useful the surrogate loss consistency framework can be. Due to how autoregressive modeling was created, the proof for Proposition 6 follows very easily once we have set up our framework. If one wanted to directly show the same result for other proper losses, it would most likely be much harder. With this framework we are able to show the result for the one loss function where it is easy and have it apply to all other proper losses.

## 6 Conclusion and future work

In this paper we explored the interplay between next-token prediction and the decoding algorithms on the one hand and different end goals on the other. Adopting an asymptotic viewpoint, we find that many of the decoding algorithms explored are not consistent for a vast majority of goals and probability distributions. Further, we find evidence that we might be asking too much of our the decoding algorithms.

Each goal we explore has one class (deterministic vs. stochastic) of decoding algorithms that, except for degenerate cases, are not consistent. What is interesting is that they flip depending on the goal. The N-gram Hamming loss can be thought of as trying to be "correct"; you need to always guess the right answer if possible, hence why randomness hurts. However, when trying to mimic a distribution, this randomness is necessary. A practical implication of our insights is that the user intent should determine the decoding strategy to be used at test time.

We believe there is a lot of interesting future work to be done in this area. One can go deeper into theoretical analysis for these or other loss functions. We particularly found theoretical analysis of the N-gram Hamming loss to be difficult due to the limited methods that can be used to manipulate indicator functions. Future work could also make use of more domain specific assumptions on probability distributions, such as power laws. Other work can include investigating other decoding algorithms, such as Top-K or nucleus sampling, or go even deeper into a specific decoder, such as temperature-scaled random sampling. One can also make the role of stochastic gradient descent more explicit in the training of next-token prediction and investigate if there are any differences that this could cause.

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

## A   Notes

### A.1   Simulation study

To investigate the optimality of $K_T$-lookahead decoding, we run a simple simulation study. The methodology went as follows:

1. Create a probability distribution over an alphabet. We do this by creating a Markov chain with $m$ nodes and our sequences are a $L$ length paths along this chain.

   (a) We create this Markov chain by having a starting distribution and its transition probabilities for each node be Dirichlet$(\alpha_1, \ldots \alpha_n)$ distributed where $\alpha_1 = \cdots = \alpha_n = \alpha$.

2. Once a graph is created, for values of $L \in \{2, 4, 6, 8\}$, $N, K \in \{1, 2, 4, 6, 8\}$, $N, K \leq L$, we find which $L$ length path is optimal for the Markov chain (representing the optimal sequence) and then see if our $K_T$-lookahead algorithm finds the optimal case.

3. For $\alpha \in \{.1, .25, .5, .75, 1, 10\}$ and $m \in \{2, 4, 6, 8\}$, do the above steps for each $(\alpha, m)$ pair 200 times and group them based on $(\alpha, m)$.

4. For each grouping, calculate the average KL-divergence of the sequence distribution with respect to the uniform distribution and calculate the average amount of times $K_T$-lookahead was $N$-gram Hamming optimal for length $L$. This is what is shown in the plots.

We chose the maximum amount of nodes $m$ and sequence length $L$ to be 8 as there are $8^8$ different sequences at the maximum (about 16 million). We show in Corollary 4.2.1 that there is no polynomial-time optimal decoder, thus we resort to brute force to go through all the combinations. Even with the use of a GPU, this still takes about 11 hours since we try every different $\alpha, m, N, K, L$ combinations described above. We also sped up computation by using memoization, but due to the exponential nature of increasing nodes or sequence length, we would start to run into memory issues if we made the length or amount of nodes larger.

The CPU used to run the simulations was an Intel 9th generation i7 and the GPU was an NVIDIA Geforce GTX 1660 Ti. The code itself was written in Python. The only non-standard package used was Numba (Lam et al., 2015). This package allows for Python code to be compiled, allowing for better wall runtime for our code. We also use its CUDA support to be able to interact with our GPU. The code for the simulations will be on Github and the link will be here if the paper is accepted. It is not here now for anonymity reasons.

For ties, we were unable to come up with a setting that would never have ties in the $K_T$-lookahead arg max. These ties come from if there is a reordering of the maximum path such that each node still proceeds to the same next node, they just do so in a different order (e.g. 15717 would output the same probability as 17157). We also note that floating point multiplication is non-associative as well, which would also make there be no ties when there should be in some cases. The way we try to remedy this is by rounding $g(y)$ for every output $y$ to 15 decimal places and then choosing the arg max over those. If $K_T$-lookahead outputted a path that was in the arg max, we considered it optimal.

We know that $K_1$-lookahead when $K = N = L$ should always produce the correct result. Thus, we thought it reasonable to see the impact of these ties by counting how many incorrect sequences were found by $K_1$-lookahead when $K = L = N$ on 200 trials of the simulation above. We found that $K_1$-lookahead was at most 3.5% unoptimal over all $\alpha$s, number of nodes, and values of $K = N = L$. All figures in this paper are accounting for ties as described in the previous paragraph.

In Figure 3 we give the a set of plots that shows $K_1$-lookahead optimality over all parameters in the simulation.

From this, we can see that the larger $N$ gets, generally the better our $K_1$-lookahead does. Since $K_T$-lookahead is choosing to walk along the path of maximum probability for each iteration, it makes sense that larger $N$s would reward this as the $N$-gram Hamming loss gets closer and closer to the $0 - 1$ loss. Smaller $N$s are more concerned with the marginal probabilities at each index, something that $K_T$-lookahead does not directly concern itself with.

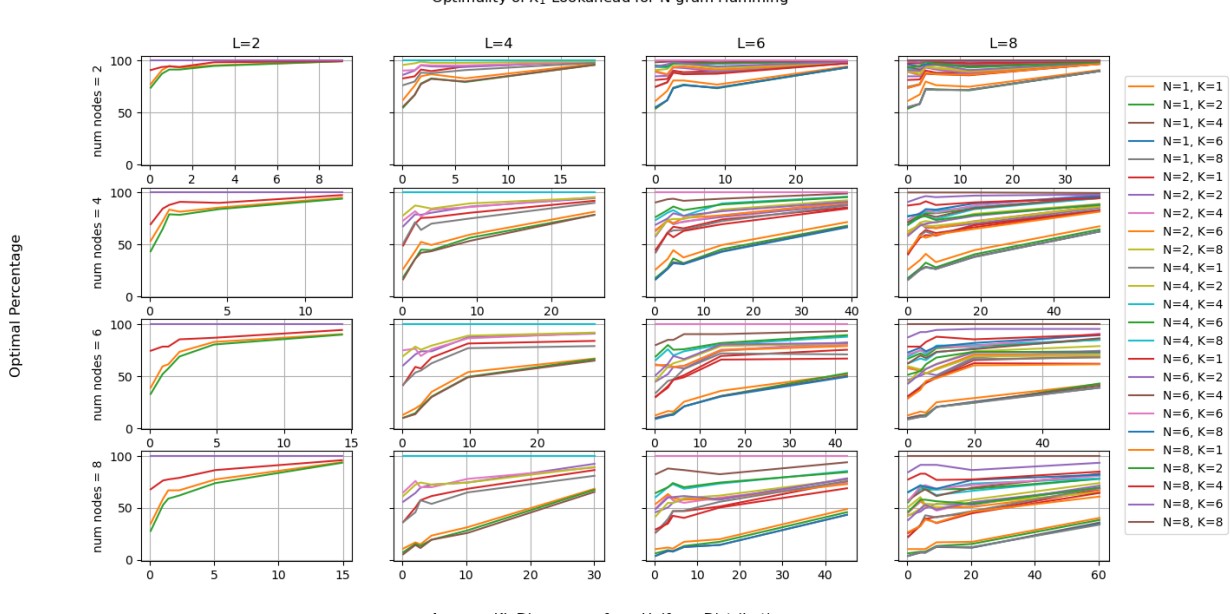

Figure 3: A plot of all trials of $K_1$-lookahead decoding being optimal for $N$-gram Hamming on a Markov chain with "num nodes" amount of nodes. The amount of nodes increases as one looks down the rows and the sequence length increases as one goes down the columns. Each line represents a specific $K_1$-lookahead being compared to the optimal $N$-gram Hamming. The $x$ and $y$ axes denote the same that has been used in Figures 1 and 2.

For completeness, we give similar plots seen for $K_1$-lookahead, but for $K_K$-lookahead in Figures 4, 5, and 6.

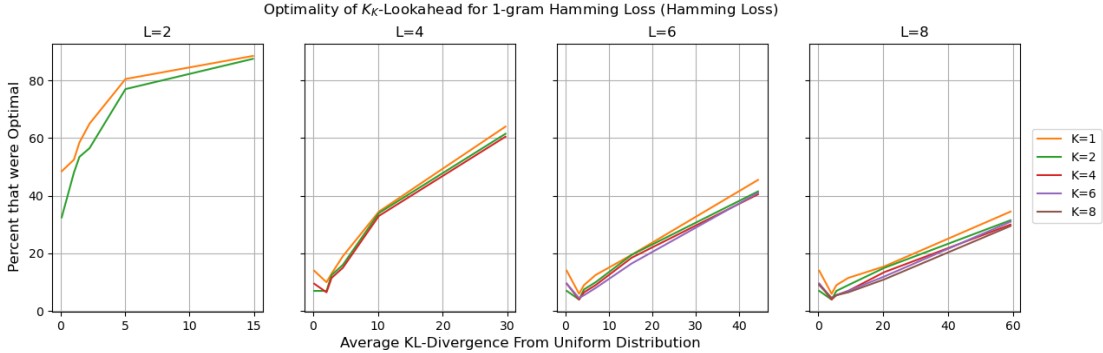

Figure 4: A plot of the amount of trials $K_K$-lookahead was optimal for the 1-gram Hamming loss (the Hamming loss). There were 8 nodes in each Markov chain and the sequence length goes up by two as one moves right in the plots.

We can see that every empirical claim made with the $K_1$-lookahead plots can also be made here. Further, we see that $K_1$ and $K_K$-lookahead both output similar looking plots, with a possible slight edge towards $K_1$ in optimality. We do not make any claims that one is better than the other. For any $T_1, T_2$ where $T_1 < T_2$, $K_{T_1}$ is going to require more compute time that $K_{T_2}$. Even though the probability distributions studied here are quite simple, a further investigation into if this extra computation is needed would be interesting given the similarity of optimality plots between $K_1$ and $K_K$-lookahead decoding.

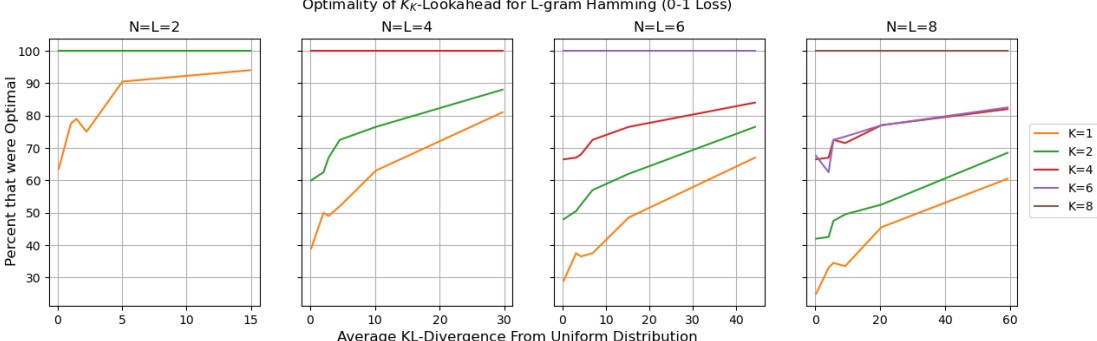

Figure 5: A plot of the amount of trials $K_K$-lookahead was optimal for the $L$-gram Hamming loss (the $0-1$ loss). The same setup as Figure 4 otherwise.

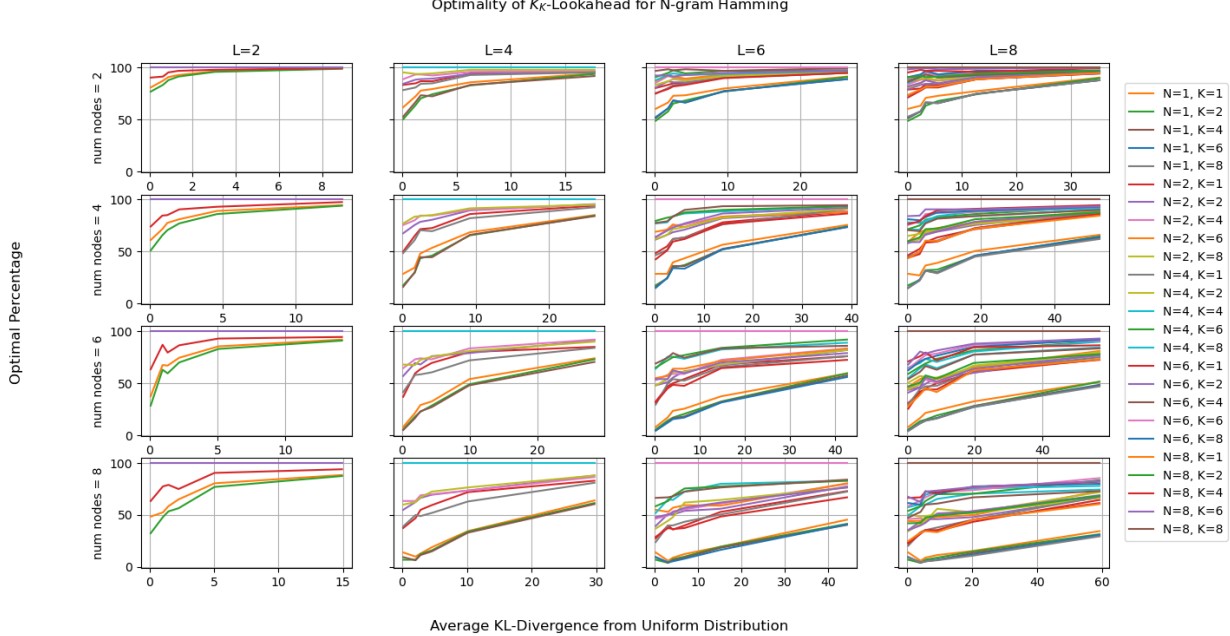

Figure 6: A plot of all trials of $K_K$-lookahead decoding being optimal for $N$-gram Hamming on a Markov chain with "num nodes" amount of nodes. The amount of nodes increases as one looks down the rows and the sequence length increases as one goes down the columns. Each line represents a specific $K_K$-lookahead being compared to the optimal $N$-gram Hamming. The $x$ and $y$ axes denote the same that has been used in Figures 4 and 5.

# B Proofs and examples

## B.1 Proof that temperature scaling is equivalent to our formulation

Let $\gamma = 1/T$ and let the set $Z$ be our logits. Then, we have that

$$p(y_i \mid y_{[i-1]}) = \frac{e^{z_i}}{\sum_{z_j \in Z} e^{z_j}}$$

Then, we have

$$\frac{p(y_i \mid y_{[i-1]})^\gamma}{\sum_{v \in \mathcal{V}} p(v \mid y_{[i-1]})^\gamma} = \frac{\dfrac{e^{z_i/T}}{\left(\sum_{z_j \in Z} e^{z_j}\right)^{1/T}}}{\sum_{z_r \in Z} \dfrac{e^{z_r/T}}{\left(\sum_{z_j \in Z} e^{z_j}\right)^{1/T}}} =$$

$$\frac{\dfrac{1}{\left(\sum_{z_j \in Z} e^{z_j}\right)^{1/T}} e^{z_i/T}}{\dfrac{1}{\left(\sum_{z_j \in Z} e^{z_j}\right)^{1/T}} \sum_{z_r \in Z} e^{z_r/T}} = \frac{e^{z_i/T}}{\sum_{z_r \in Z} e^{z_r/T}}$$

This last value is how temperature scaling is implemented.

We do note this $T$ is different than the $T$ used in the rest of the paper for $K_T$-lookahead. We use $T$ here to represent the softmax temperature as it is the standard notation for it, but nowhere else in this paper do we use it to represent temperature.

## B.2 Proof of assertion in Assumption 3.1

**Lemma B.1.** *Suppose we have $\forall i \in [L]$, $\forall y_{[i]} \in \mathcal{Y}_{[i]}$, and $\forall v \in \mathcal{V}$,*

$$p^i_{ntp}(v \mid y_{[i]}) \rightarrow p^*(v \mid y_{[i]}).$$

*Then $KL(p^* \| p^i_{ntp}) \rightarrow 0$.*

*Proof.* Below we begin by expanding the KL-divergence out and using expectation and log properties to decompose it into a function of the conditional KL-Divergences.

$$KL(p^* \| p^i_{ntp}) = \underset{y \sim p^*}{\mathbb{E}} \left[ -\log \left( \frac{p^i_{ntp}(y)}{p(y)} \right) \right] = \sum_{j=1}^{L} \underset{y \sim p^*}{\mathbb{E}} \left[ -\log \left( \frac{p^i_{ntp}(y_j \mid y_{[j-1]})}{p(y_j \mid y_{[j-1]})} \right) \right] =$$

$$\sum_{j=1}^{L} \sum_{y \in \mathcal{Y}} -p^*(y) \log \left( \frac{p^i_{ntp}(y_j \mid y_{[j-1]})}{p(y_j \mid y_{[j-1]})} \right) =$$

$$\sum_{j=1}^{L} \sum_{y \in \mathcal{Y}} -p^*(y_{[j-1]}) p^*(y_j \mid y_{[j-1]}) p^*(y_{j+1:} \mid y_{[j]}) \log \left( \frac{p^i_{ntp}(y_j \mid y_{[j-1]})}{p(y_j \mid y_{[j-1]})} \right) \qquad (\star)$$

Let us create, for each $i \in [L]$:

$$Y_{[i]} = \{ y_{[i]} \mid y \in \mathcal{Y} \},$$
$$Y_{i+1:} = \{ y_{i+1:} \mid y \in \mathcal{Y} \}.$$

We can see the inner sum then becomes:

$$\sum_{y_{[j]} \in \mathcal{Y}_{[j]}} \sum_{y_{j+1:} \in \mathcal{Y}_{j+1:}} -p^*(y_{[j-1]}) p^*(y_j \mid y_{[j-1]}) p^*(y_{j+1:} \mid y_{[j]}) \log \left( \frac{p^i_{ntp}(y_j \mid y_{[j-1]})}{p(y_j \mid y_{[j-1]})} \right) =$$

$$\sum_{y_{[j]} \in \mathcal{Y}_{[j]}} -p^*(y_{[j-1]}) p^*(y_j \mid y_{[j-1]}) \log \left( \frac{p^i_{ntp}(y_j \mid y_{[j-1]})}{p(y_j \mid y_{[j-1]})} \right) \sum_{y_{j+1:} \in \mathcal{Y}_{j+1:}} p^*(y_{j+1:} \mid y_{[j]}).$$

From the definition of $\mathcal{Y}_{j+1:}$, we have that this last sum is 1. Thus, $(\star)$ becomes:

$$\sum_{j=1}^{L} \sum_{y_{[j]} \in \mathcal{Y}_{[j]}} -p^*(y_{[j-1]}) p^*(y_j \mid y_{[j-1]}) \log \left( \frac{p_{ntp}^i(y_j \mid y_{[j-1]})}{p(y_j \mid y_{[j-1]})} \right) =$$

$$\sum_{j=1}^{L} \sum_{y_{[j-1]} \in \mathcal{Y}_{[j-1]}} \sum_{v \in \mathcal{V}} -p^*(y_{[j-1]}) p^*(v \mid y_{[j-1]}) \log \left( \frac{p_{ntp}^i(v \mid y_{[j-1]})}{p(v \mid y_{[j-1]})} \right) =$$

$$\sum_{j=1}^{L} \sum_{y_{[j-1]} \in \mathcal{Y}_{[j-1]}} p^*(y_{[j-1]}) \sum_{v \in \mathcal{V}} -p^*(v \mid y_{[j-1]}) \log \left( \frac{p_{ntp}^i(v \mid y_{[j-1]})}{p(v \mid y_{[j-1]})} \right) =$$

$$\sum_{j=1}^{L} \sum_{y_{[j-1]} \in \mathcal{Y}_{[j-1]}} p^*(y_{[j-1]}) \mathop{\mathbb{E}}_{v \sim p^*(\cdot \mid y_{[j-1]})} \left[ -\log \left( \frac{p_{ntp}^i(v \mid y_{[j-1]})}{p(v \mid y_{[j-1]})} \right) \right].$$

By assumption, we have that

$$\mathop{\mathbb{E}}_{v \sim p^*(\cdot \mid y_{[j-1]})} \left[ -\log \left( \frac{p_{ntp}^i(v \mid y_{[j-1]})}{p(v \mid y_{[j-1]})} \right) \right] \to 0$$

for each one. Thus, since there are a finite number of terms, standard arguments show that that the entire function will limit to 0. $\qquad\square$

### B.3 Proof of Proposition 1

*Proof.* We will show that the limit of the difference of the expected risks is 0.

$$\mathop{\mathbb{E}}_{x \sim p_x, y \sim p^* \mid x, \hat{y} \sim p_{\mathcal{D}(p_{ntp}^i) \mid x}} [\ell(\hat{y}, y)] - \mathop{\mathbb{E}}_{x \sim p_x, y \sim p^* \mid x, \hat{y} \sim p_{\mathcal{D}(p_{ntp}^*) \mid x}} [\ell(\hat{y}, y)] =$$

$$\mathop{\mathbb{E}}_{x \sim p_x, y \sim p^* \mid x} \left[ \sum_{\hat{y} \in \mathcal{Y}} \left( p_{\mathcal{D}(p_{ntp}^i) \mid x}(\hat{y}) - p_{\mathcal{D}(p_{ntp}^*) \mid x}(\hat{y}) \right) \ell(\hat{y}, y) \right] =$$

$$\mathop{\mathbb{E}}_{x \sim p_x} \left[ \sum_{\hat{y} \in \mathcal{Y}} \left( p_{\mathcal{D}(p_{ntp}^i) \mid x}(\hat{y}) - p_{\mathcal{D}(p_{ntp}^*) \mid x}(\hat{y}) \right) \mathop{\mathbb{E}}_{y \sim p^* \mid x} [\ell(\hat{y}, y)] \right] =$$

$$\sum_{\hat{y} \in \mathcal{Y}} \mathop{\mathbb{E}}_{x \sim p_x} \left[ \left( p_{\mathcal{D}(p_{ntp}^i) \mid x}(\hat{y}) - p_{\mathcal{D}(p_{ntp}^*) \mid x}(\hat{y}) \right) \mathop{\mathbb{E}}_{y \sim p^* \mid x} [\ell(\hat{y}, y)] \right].$$

Now, by assumption we have

$$p_{\mathcal{D}(p_{ntp}^i) \mid x}(\hat{y}) - p_{\mathcal{D}(p_{ntp}^*) \mid x}(\hat{y}) \to 0$$

and notice that $\left| \left( p_{\mathcal{D}(p_{ntp}^i) \mid x}(\hat{y}) - p_{\mathcal{D}(p_{ntp}^*) \mid x}(\hat{y}) \right) \mathbb{E}_{y \sim p^* \mid x} [\ell(\hat{y}, y)] \right| \leq M$. Thus, by the dominated convergence theorem, we have

$$\lim_{i \to \infty} \mathop{\mathbb{E}}_{x \sim p_x} \left[ \left( p_{\mathcal{D}(p_{ntp}^i) \mid x}(\hat{y}) - p_{\mathcal{D}(p_{ntp}^*) \mid x}(\hat{y}) \right) \mathop{\mathbb{E}}_{y \sim p^* \mid x} [\ell(\hat{y}, y)] \right] = \mathop{\mathbb{E}}_{x \sim p_x} \left[ \lim_{i \to \infty} \left( p_{\mathcal{D}(p_{ntp}^i) \mid x}(\hat{y}) - p_{\mathcal{D}(p_{ntp}^*) \mid x}(\hat{y}) \right) \mathop{\mathbb{E}}_{y \sim p^* \mid x} [\ell(\hat{y}, y)] \right] = 0.$$

Thus, since $|\hat{\mathcal{Y}}| < \infty$, we have

$$\lim_{i \to \infty} \sum_{\hat{y} \in \mathcal{Y}} \mathop{\mathbb{E}}_{x \sim p_x} \left[ \left( p_{\mathcal{D}(p_{ntp}^i) \mid x}(\hat{y}) - p_{\mathcal{D}(p_{ntp}^*) \mid x}(\hat{y}) \right) \mathop{\mathbb{E}}_{y \sim p^* \mid x} [\ell(\hat{y}, y)] \right] =$$

$$\sum_{\hat{y} \in \mathcal{Y}} \lim_{i \to \infty} \mathop{\mathbb{E}}_{x \sim p_x} \left[ \left( p_{\mathcal{D}(p_{ntp}^i) \mid x}(\hat{y}) - p_{\mathcal{D}(p_{ntp}^*) \mid x}(\hat{y}) \right) \mathop{\mathbb{E}}_{y \sim p^* \mid x} [\ell(\hat{y}, y)] \right] = 0,$$

which is what we needed to show $\qquad\square$

## B.4 Proof of Lemma 4.1

*Proof.* Let $\hat{y}$ be the output of our algorithm. By linearity of expectation, we have

$$\mathbb{E}\left[\sum_{i=1}^{L-N+1} \mathbf{1}_{\{\hat{y}_{i:i+N-1} \neq y_{i:i+N-1}\}}\right] = \sum_{i=1}^{L-N+1} \mathbb{E}\left[\mathbf{1}_{\{\hat{y}_{i:i+N-1} \neq y_{i:i+N-1}\}}\right]$$

We can see that

$$\mathbb{E}\left[\mathbf{1}_{\{\hat{y}_{i:i+N-1} \neq y_{i:i+N-1}\}}\right] = 1 - p(\hat{y}_{i:i+N-1})$$

Therefore, we have

$$\mathbb{E}\left[\sum_{i=1}^{L-N+1} \mathbf{1}_{\{\hat{y}_{i:i+N-1} \neq y_{i:i+N-1}\}}\right] = \sum_{i=1}^{L-N+1} 1 - p(\hat{y}_{i:i+N-1}) =$$

$$L - N + 1 - \sum_{i=1}^{L-N+1} p(\hat{y}_{i:i+N-1}) = L - N + 1 - g(\hat{y})$$

Therefore, maximizing $g(y)$ will minimize our expected risk. $\qquad\square$

## B.5 Proof of Theorem 4.2

*Proof.* We have that

$$\forall y \in \mathcal{Y}, \quad p(y) = p(y_1)p(y_2|y_1)\dots p(y_L|y_{[L-1]}).$$

We can think of this as a path $y_1 \to y_2 \to \cdots \to y_L$. We can combine all these paths to make a directed tree. Let each node have the weight of the conditional distribution at that node. Thus, if we take the product of any path $y_1 \to y_2 \to \cdots \to y_L$, we get $p(y)$.

Suppose $p$ is defined as in the Theorem 4.2 and suppose we do not query $|\mathcal{V}|^L - 1$ conditional probability values. We note that $|\mathcal{V}|^L - 1 = \frac{|\mathcal{V}|-1}{|\mathcal{V}|} \sum_{j=1}^{L} |\mathcal{V}|^j$. We know that the $j^{\text{th}}$ level in our tree has $|\mathcal{V}|^j$ nodes. Therefore, on at least one level, the ratio of nodes we have queried is less than $\frac{|\mathcal{V}|-1}{|\mathcal{V}|}$. Thus, by the pigeonhole principle, there must exist two nodes that have the same path up until that point, $v|y_{[j-1]}$, $u|y_{[j-1]}$, that have not been looked at. Therefore, the algorithm is unable to know the exact probability of any descendants of these two paths. If either of these nodes have a weight more than $1/|\mathcal{V}|$, say without loss of generality it is $v|y_{[j-1]}$, then $g(y_{[j-1]} + v + \dots)$ would be larger than any path $\mathcal{D}$ has found so far. Therefore, the algorithm can not be sure either answer is optimal and must query more. Thus, since the algorithm was arbitrary, on this distribution any algorithm runtime will be at least $C(|\mathcal{V}|^L - 1)$. $\qquad\square$

## B.6 Proof of Lemma 4.3

*Proof.* Let $x \in \mathcal{X}$, $k \in \mathbb{N}$ and let $p^i|x \to p^*|x$. Then, let

$$\epsilon_i = \min\left\{\left|p^i(y_{cK+1}, \dots, y_{cK+K} \mid y_{[cK]}, x) - p^*(y_{cK+1}, \dots, y_{cK+K} \mid y_{[cK]}, x)\right| \mid c \in \mathbb{Z}_+\right\}$$

Notice, in order for $p^i|x \to p^*|x$, we need $\epsilon_i \to 0$. Thus, since there are only a finite amount of marginal and conditional values our decoder can look at, and since we know there are no ties, there will be some $j$ such that $\forall i > j$ the $\arg\max$ for the conditional distributions of both $p^i$ and $p^*$ will match. Therefore, we meet the assumption needed to use Prop 1. $\qquad\square$

## B.7 Proof of Proposition 2

*Proof.* • $p_x^*(C) = 1 \implies K_T$-**lookahead optimality**: By the defintion of $C$, we know the $K_T$-lookahead outputs maximize $\sum_{i=1}^{L-N+1} p^*(y_{i:i+N-1} \mid x)$ except a set of measure 0 over $X$. From Lemma 4.1, we can see that this is the optimal output.

- $K_T$-**lookahead optimality** $\implies p_x^*(C) = 1$: We will prove the contraposition. Suppose $p_x^*(C) < 1$. Then, there exists a set $L \in \mathcal{X}$ of measure $> 0$ where for each $x \in L$:

$$\arg \max_y \sum_{i=1}^{L-N+1} p^*(y_{i:i+N-1} \mid x) = y^\dagger,$$

but

$$y^\dagger \neq \hat{y},$$

where $\hat{y}$ what our $K_T$-lookahead decoding algorithm outputs. Since we know $y^\dagger$ is optimal, K-lookahead decoding will be unoptimal.

$\square$

## B.8 Example of a Markov chain that is not $K_T$-lookahead optimal

Let $K, L, N, T \in \mathbb{N}$ such that $T \leq K < L$ and $N \leq L$ so that this set up makes sense in the context of this paper. Let us have the following Markov chain:

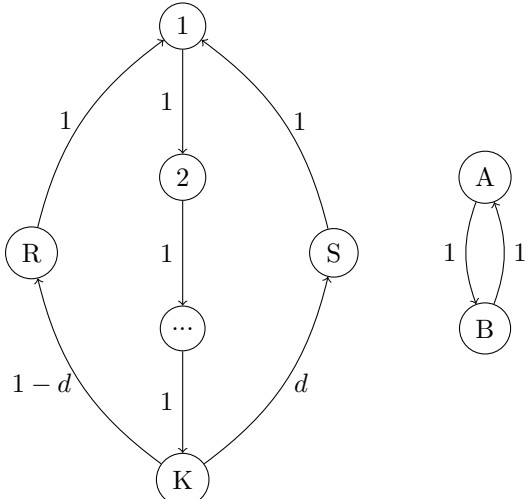

for some $\frac{1}{2} < d < 1$. Then, note how $K_T$-lookahead will repeat $123 \ldots KS$ to length $L$. Let $s = s_1 s_2 \ldots s_L$ be the output of $K_T$-lookahead and let $a = a_1 \ldots a_L$ be the sequence $ABAB \ldots$ to size $L$. We can see that that amount of $a_i$'s such that $a_i = S$ will be $\left\lfloor \frac{L}{K+1} \right\rfloor$. We can see that, since $K < L$, we have $1 \leq \left\lfloor \frac{L}{K+1} \right\rfloor$. Now, let $c$ be the amount of $N$-grams in $s$ that contain an $S$. Since $1 \leq \left\lfloor \frac{L}{K+1} \right\rfloor$, we have $1 \leq c$. Let

$$0 < \epsilon < \frac{1}{4\left(L - N + 1 - \frac{1-d}{2}c\right)}.$$

We can see that our upper bound is always greater than 0 as $c \leq L - N + 1$ and $d < 1$, therefore this $\epsilon$ is valid for any possible values of $K, T, N, L$ and $d$ that work in this setting. Let our initial distribution of the Markov chain, $\pi$, be such that $\mathbb{P}_\pi (1) = .50 + \epsilon$ and $\mathbb{P}_\pi (A) = .50 - \epsilon$. If $s_i \ldots s_{i+N-1}$ is an $N$-gram of $s$ that contains $S$, we have that

$$\mathbb{P}(s_i \ldots s_{i+N-1}) \leq d\left(\frac{1}{2} + \epsilon\right)$$

because there is at least one $S$ in it (and possibly more if $N$ is large enough). If it does not contain an $S$, then we can see that

$$\mathbb{P}(s_i \ldots s_{i+N-1}) = \frac{1}{2} + \epsilon.$$

Therefore, we have the following:

$$\sum_{n=1}^{L-N+1} \mathbb{P}\left(s_n \ldots s_{n+N-1}\right) \le cd\left(\frac{1}{2}+\epsilon\right) + \sum_{n=1}^{L-N+1-c}\left(\frac{1}{2}+\epsilon\right) = cd\left(\frac{1}{2}+\epsilon\right) + (L-N+1-c)\left(\frac{1}{2}+\epsilon\right).$$

We can also see that for the sequence $a$:

$$\sum_{n=1}^{L-N+1} \mathbb{P}\left(a_n \ldots a_{n+N-1}\right) = (L-N+1)\left(\frac{1}{2}-\epsilon\right).$$

In order for $K_T$-lookahead to not be optimal, we need

$$(L-N+1)\left(\frac{1}{2}-\epsilon\right) > cd\left(\frac{1}{2}+\epsilon\right) + (L-N+1-c)\left(\frac{1}{2}+\epsilon\right).$$

From this, below we manipulate algebra to arrive that $\epsilon$ must be smaller than its upper bound.

$$(L-N+1)\left(\frac{1}{2}-\epsilon\right) > cd\left(\frac{1}{2}+\epsilon\right) + (L-N+1-c)\left(\frac{1}{2}+\epsilon\right) \implies$$

$$\frac{1}{2}(L-N+1) - \epsilon(L-N+1) > \frac{1}{2}(L-N+1-(1-d)c) + \epsilon(L-N+1-(1-d)c) \implies$$

$$\frac{1}{2}(L-N+1-L+N-1+(1-d)c) > \epsilon(L-N+1+L-N+1-(1-d)c) \implies$$

$$\epsilon < \frac{1}{2}\frac{(1-d)c}{2\left(L-N+1-\frac{1-d}{2}c\right)} = \frac{(1-d)c}{4\left(L-N+1-\frac{1-d}{2}c\right)}$$

Therefore, we have that $a$ is a more optimal sequence for this Markov chain/initial distribution pair than the output of $K_T$-lookahead, which is what we wanted to show.

### B.9 Proof of Proposition 3

*Proof.* Let $N < L$. Then, let us have the following probability distribution:

$$p(0\ldots0) = .28, \quad p(\underbrace{0\ldots0}_{K_2-1 \text{ indices}} 10\ldots0) = .12, \quad p(20\ldots0) = .23, \quad p(11\ldots1) = .37.$$

One can verify that the $K_{1_{T_1}}$-lookahead decoder would output $0\ldots0$ and the $K_{2_{T_2}}$-lookahead decoder would output $1\ldots1$. One can also verify that for any N-gram starting at position $c > 1$,

$$\underbrace{0\ldots0}_{N \text{ indices}} = \arg\max_{y_{c:c+N-1}} p(y_{c:c+N-1})$$

since $p(0_j \ldots 0_{j+N-1}) \ge .28 + .23 = .51$ where $j > 1$. From this, by calculating the conditional and marginal distributions for the first N-gram, one can see that $\arg\max_{y\in\mathcal{Y}}\{g(y)\} = 0\ldots0$. Therefore, the $K_{1_{T_1}}$-lookahead decoder is optimal for the n-gram Hamming loss, while the $K_{2_{T_2}}$-lookahead decoder is not.

For $N = L$, we have the $0-1$ loss, whose optimal output is the max probability sequence. let us have the following probability distribution:

$$p(0\ldots0) = .408, \quad p(\underbrace{0\ldots0}_{K_2-1 \text{ indices}} 11\ldots1) = .102,$$

$$p(\underbrace{1\ldots1}_{K_2 \text{ indices}} 00\ldots0) = .2401, \quad p(11\ldots1) = .2499.$$

Here, we can see that the $K_{1_{T_1}}$-lookahead decoder will output $0\ldots0$, however, since the max marginal for the first $K_2$ is $1\ldots1$, the $K_{2_{T_2}}$-lookahead decoder will not output $0\ldots0$.

Thus, we have covered all cases and have shown what was need. $\qquad\square$

## B.10 Proof of Proposition 4 and monotonicity result

*Proof.* Now, let $N, K, L, T$ be as stated in Proposition 4. We will constructively create a two counterexamples, one for when $N < L$ and another for when $N = L$. Let our alphabet be $\{0, 1, 2\}$. For the $N < L$ case, we have the following probability distribution:

$$p(0\ldots 0) = .27675, \quad p(10\ldots 0) = .25, \quad p(\underbrace{0\ldots 0}_{K \text{ indices}} 20\ldots 0) = .03075,$$

$$p(\underbrace{0\ldots 0}_{T \text{ indices}} 1\ldots 1) = .2925, \quad p(\underbrace{0\ldots 0}_{T \text{ indices}} 20\ldots 0) = .15.$$

It is easy to see that both $K_T$ and $K_{T+1}$ will both choose $0\ldots 0$ for their first $T$ and $T + 1$ values respectively. From this, we can see that this locks in $T + 1$ into choosing rather $0\ldots 0$ or $\underbrace{0\ldots 0}_{T \text{ indices}} 20\ldots 0$, from which one can see it will choose $0\ldots 0$ by following the algorithm. For $K_T$, it sees the following for its second iteration:

$$p(\underbrace{1\ldots 1}_{K \text{ indices}} \mid \underbrace{0\ldots 0}_{T \text{ indices}}) = .39, \quad p(\underbrace{0\ldots 0}_{K-T \text{ indices}} \underbrace{20\ldots 0}_{T \text{ indices}} \mid \underbrace{0\ldots 0}_{T \text{ indices}}) = .041, \quad p(\underbrace{0\ldots 0}_{K \text{ indices}} \mid \underbrace{0\ldots 0}_{T \text{ indices}}) = .369,$$

$$p(\underbrace{20\ldots 0}_{K \text{ indices}} \mid \underbrace{0\ldots 0}_{T \text{ indices}}) = .2.$$

From this, we can see that it will choose $\underbrace{1\ldots 1}_{T \text{ indices}}$ and then be locked into the sequence $\underbrace{0\ldots 0}_{T \text{ indices}} 1\ldots 1$. Now that we know both of the outputs of $K_T$ and $K_{T+1}$, we need to show that $0\ldots 0$ is optimal. Notice that for any N-gram starting at position $c > 1$:

$$p(0_c\ldots 0_{c+N-1}) \geq .27675 + .25 = .52675$$

and there are no ties in the arg max. Let us then have a sequence $y$. Let $S_y$ be all $N$-grams of $y$ that contain a non-0 index. Notice that:

$$\forall y_{j:j+N-1} \in S_{y_{1:}} \quad p(y_{j:j+N-1}) < p(0_j \ldots 0_{j+N-1}).$$

Thus, for every starting index greater than 1, our sequence would be better off it was only 0s. Therefore, we only need to show the same for index 1. By calculating the marginal and conditional distributions, it can be seen that, for every $N$, $p(0_1\ldots 0_N) > p(y_{1:N-1}) - .02$ where $y_{1:N-1}$ is any $N$-gram that is not all zeros. Therefore, since for any $c > 1$, $p(0_c\ldots 0_{c+N-1}) > .5 + .02$, we know that

$$g(0\ldots 0) = \sum_{i=1}^{L-N+1} p(0_i\ldots 0_{i+N-1}) > \sum_{i=1}^{L-N+1} p(y_i\ldots y_{i+N-1}) = g(y)$$

for any other sequence $y$. Thus $0\ldots 0$ is optimal.

For $N = L$, let our alphabet be $\{0, 1, 2\}$. Now, we will define marginal and conditional probabilities for the first $K$ indices for the probability distribution:

$$p(\underbrace{0\ldots 0}_{T \text{indices}}) = 1, \quad p(\underbrace{0\ldots 0}_{K-T \text{ indices}} \mid \underbrace{0\ldots 0}_{T \text{indices}}) = .51, \quad p(\underbrace{1\ldots 1}_{K-T \text{ indices}} \mid \underbrace{0\ldots 0}_{T \text{indices}}) = .49.$$

From this, we can see that $K_{T+1}$ will choose $\underbrace{0\ldots 0}_{T+1 \text{ indices}}$ for the first round but $K_T$ only chooses $\underbrace{0\ldots 0}_{T \text{ indices}}$. The goal now is to adversarially create the rest of the sequence probabilities so that $K_T$ and $K_{T+1}$ diverge and $K_{T+1}$ is optimal. Let us now give the full probability distribution:

$$p(\underbrace{0\ldots 0}_{K \text{ indices}} 2\ldots 2) = .051, \quad p(0\ldots 0) = .459,$$

$$p(\underbrace{0\ldots 0}_{T \text{ indices}} 1\ldots 1) = .2499, \quad p(\underbrace{0\ldots 0}_{T \text{ indices}} \underbrace{1\ldots 1}_{K \text{ indices}} 0\ldots 0) = .2401.$$

We note that since $K < L - T \implies K + T < L$, the last two sequences above are distinct (i.e., there is at least one 0 at the end of the last sequence). Notice how we have created two paths that diverge at the $T+1$ spot depending on if the $T+1$ spot is a 0 or a 1. On the second iteration, $K_T$ will see that $p(\underbrace{1\ldots1}_{K \text{ indices}} \mid \underbrace{0\ldots0}_{T \text{ indices}}) = .49$, while any other choices would have less probability than that, thus we have that $K_T$ will choose 1 at the $T+1$ spot. Since $K_{T+1}$ already chose a 0 at that spot, their paths have split. Specifically, we can see that $K_T$ will choose $\underbrace{0\ldots0}_{T \text{ indices}} 1\ldots1 = \hat{y}$ and $K_{T+1}$ will choose $0\ldots0 = y^\dagger$. Since $N = L$, we know the optimal sequence is the one with the most probability, which is $0\ldots0$, which shows what we needed. □

For the monotonicity result, let $K \in \{2, 3\ldots, L\}$, $N = L$, $T_1, T_2 \in [K]$ such that $T_1 < T_2$. Suppose also $K \geq L - T_1$. $K_{T_1}$ and $K_{T_2}$ are looking over the same $K$ tokens in the first iteration, thus their first $T_1$ values will be the same. Then, since $K \geq L - T_1$, we know $K_{T_1}$ will choose the optimal rest of the tokens since it looks over every possibility left. Therefore, we only need to know if its first $T_1$ tokens were optimal. Since $K_{T_2}$ is optimal, and they share the same first $T_1$ tokens, we then know that $K_{T_1}$ is optimal.

### B.11 Proof of Proposition 5

*Proof.* Let $p$ be our probability distribution over $\mathcal{X} \times \mathcal{Y}$ and let $\mathcal{D}$ be our decoding algorithm. By lemma 4.1, given an input $x$, the optimal output is $\arg\max_y g(y|x)$. Notice:

$$\mathop{\mathbb{E}}_{y \sim p|x, \hat{y} \sim p_{\mathcal{D}(p_{ntp})|x}} \left[ \sum_{i=1}^{L-N+1} \mathbf{1}_{\{y_{i:i+N-1} \neq \hat{y}_{i:i+N-1}\}} \right] =$$

$$\sum_{\hat{y} \in \mathcal{Y}} p_{\mathcal{D}(p_{ntp})|x}(\hat{y}) \mathop{\mathbb{E}}_{y \sim p|x} \left[ \sum_{i=1}^{L-N+1} \mathbf{1}_{\{y_{i:i+N-1} \neq \hat{y}_{i:i+N-1}\}} \right].$$

We know that $\sum_{\hat{y} \in \mathcal{Y}} p_{\mathcal{D}(p_{ntp})|x}(\hat{y}) = 1$. Therefore, in order to minimize our total sum, we need all the mass of $p_{\mathcal{D}(p_{ntp})|x}(\hat{y})$ to be on values of $\hat{y}$ which minimize our expected risk. Since this was for an arbitrary $x \in \mathcal{X}$, we have shown what was needed. □

### B.12 Random sampling and temperature-scaled random sampling meet the assumption needed for Proposition 1

Let us look at one particular $y$. Let $p_{RS(p|x)}(\cdot)$ be the probability distribution of random sampling decoder using $p$ as a next-token predictor given an input $x$ and $p_{TSRS(p|x,\gamma)}(\cdot)$ be the same for temperature scaled random sampling with hyperparameter $\gamma$. By defintion, we have for every $x \in \mathcal{X}$ and $y \in \mathcal{Y}$:

$$p_{RS(p|x)}(y) = \prod_{i=1}^{L} p(y_i \mid y_{[i-1]}, x)$$

$$p_{TSRS(p|x,\gamma)}(y) = \prod_{i=1}^{L} \frac{p(y_i \mid y_{[i-1]}, x)^\gamma}{\sum_{v \in \mathcal{V}} p(v \mid y_{[i-1]}, x)^\gamma}.$$

We can see that each of these are continuous in $p(\cdot \mid \cdot)$ so long as $\gamma \neq \infty$. Thus, as $p^i \to p^*$, we have that

$$p_{RS(p^i|x)}(y) \to p_{RS(p^*|x)}(y)$$

and

$$p_{TSRS(p^i|x,\gamma)}(y) \to p_{TSRS(p^*|x,\gamma)}(y).$$

If $\gamma = \infty$, then this becomes greedy decoding, which we show in Lemma 4.3 meets the assumption needed as well.

### B.13 Proof of Proposition 6

*Proof.* By the probability chain rule, we can see that random sampling from $p^i_{ntp}(\cdot \mid \cdot)$ and then concatenating has the same distribution as sampling from $p^i$ itself. Therefore, we will work with $p^i$ for the rest of the proof without regard for next-token prediction. Given that $H(p)$ is the entropy of a probability distribution $p$, we have

$$CE(p^i, p^*) =$$
$$\mathbb{E}_{y \sim p^*}\left[-\log\left(p^i(y)\right)\right] = \mathbb{E}_{y \sim p^*}\left[-\log\left(p^i(y)\right)\right] + \mathbb{E}_{y \sim p^*}\left[-\log\left(p^*(y)\right)\right] - \mathbb{E}_{y \sim p^*}\left[-\log\left(p^*(y)\right)\right] =$$
$$\mathbb{E}_{y \sim p^*}\left[-\log\left(\frac{p^i(y)}{p^*(y)}\right)\right] + \mathbb{E}_{y \sim p^*}\left[-\log\left(p^*(y)\right)\right] = KL(p^* || p^i) + H(p^*).$$

By Assumption 3.1 we have that $p^i \to p^*$ in KL-Divergence. Since KL-Divergence is also a metric, we have that $CE(p^i, p^*) \geq H(p^*)$. Thus, we can see that $\lim_{i \to \infty} CE(p^i, p^*) = H(p^*)$, which shows we obtain the minimum value we can and therefore have consistency. □

### B.14 Proof of Proposition 7

*Proof.* We know that $p^i \to p^*$ in KL-divergence. Further, by Appendix B.12, we can see that for all $y \in \mathcal{Y}$ and $x \in \mathcal{X}$, $p_{RS(p^i_{ntp}|x)}(y) \to p_{RS(p^*_{ntp}|x)}(y)$ and $p_{TSRS(p^i_{ntp}|x,\gamma)}(y) \to p_{TSRS(p^*_{ntp}|x,\gamma)}(y)$. Thus, if we can show that $p_{TSRS(p^*_{ntp}|x,\gamma)} \neq p_{RS(p^*_{ntp}|x)}$, then we are done. Let $p$ be the limit for random sampling and let $p^\gamma$ be the limit for temperature scaled random sampling for temperature parameter $\gamma$.

Let $\gamma \neq 1$. In section 5.2 we know that random sampling is optimal. In Appendix B.13 we show the well known fact that cross entropy is the sum of the entropy of the true distribution plus the KL-Divergence of the two distributions. The KL-divergence has a unique minimum at the true distribution. Thus, we will show that $KL(p || p^\gamma) = 0$ if and only if $p$ is uniform or deterministic. We begin by using the same analysis done in Appendix B.2 to break the KL-Divergence up into a function of the conditional KL-Divergences.

$$KL(p || p^\gamma) = \mathbb{E}_{y \sim p}\left[-\log\left(\frac{p^\gamma(y)}{p(y)}\right)\right] = \sum_{i=1}^{L} \mathbb{E}_{y \sim p}\left[-\log\left(\frac{\frac{p(y_i|y_{[i-1]})^\gamma}{\sum_{v \in \mathcal{V}} p(v|y_{[i-1]})^\gamma}}{p(y \mid y_{[i-1]})}\right)\right] =$$

$$\sum_{i=1}^{L} \mathbb{E}_{y \sim p}\left[-\log\left(\frac{p(y_i \mid y_{[i-1]})^{\gamma-1}}{\sum_{v \in \mathcal{V}} p(v \mid y_{[i-1]})^\gamma}\right)\right] = \sum_{i=1}^{L} \sum_{y \in \mathcal{Y}} -p(y)\log\left(\frac{p(y_i \mid y_{[i-1]})^{\gamma-1}}{\sum_{v \in \mathcal{V}} p(v \mid y_{[i-1]})^\gamma}\right) =$$

$$\sum_{i=1}^{L} \sum_{y \in \mathcal{Y}} -p(y_{[i-1]})p(y_i \mid y_{[i-1]})p(y_{[i+1:]} \mid y_{[i]})\log\left(\frac{p(y_i \mid y_{[i-1]})^{\gamma-1}}{\sum_{v \in \mathcal{V}} p(v \mid y_{[i-1]})^\gamma}\right) =$$

$$\sum_{i=1}^{L} \sum_{y_{[i]} \in \mathcal{Y}_{[i]}} -p(y_{[i-1]})p(y_i \mid y_{[i-1]})\log\left(\frac{p(y_i \mid y_{[i-1]})^{\gamma-1}}{\sum_{v \in \mathcal{V}} p(v \mid y_{[i-1]})^\gamma}\right) =$$

$$\sum_{i=1}^{L} \sum_{y_{[i-1]} \in \mathcal{Y}_{[i-1]}} p(y_{[i-1]}) \sum_{y_i \in \mathcal{V}} -p(y_i \mid y_{[i-1]})\log\left(\frac{p(y_i \mid y_{[i-1]})^{\gamma-1}}{\sum_{v \in \mathcal{V}} p(v \mid y_{[i-1]})^\gamma}\right) =$$

$$\sum_{i=1}^{L} \sum_{y_{[i-1]} \in \mathcal{Y}_{[i-1]}} p(y_{[i-1]}) \mathbb{E}_{y_i \sim p(\cdot|y_{[i-1]})}\left[-\log\left(\frac{p(y_i \mid y_{[i-1]})^{\gamma-1}}{\sum_{v \in \mathcal{V}} p(v \mid y_{[i-1]})^\gamma}\right)\right] =$$

$$\sum_{i=1}^{L} \sum_{y_{[i-1]} \in \mathcal{Y}_{[i-1]}} p(y_{[i-1]}) \mathbb{E}_{y_i \sim p(\cdot|y_{[i-1]})}\left[-\log\left(\frac{\frac{p(y_i|y_{[i-1]})^\gamma}{\sum_{v \in \mathcal{V}} p(v|y_{[i-1]})^\gamma}}{p(y_i \mid y_{[i-1]})}\right)\right].$$

Thus, we can see that $KL(p||p^\gamma)$ is a function of the KL-divergence of the conditional probability distributions. Since we need $KL(p||p^\gamma) = 0$, this would then make us need each conditional KL-divergence also need to be 0. Thus, we require for every $y_{[i-1]} \in \mathcal{Y}_{[i-1]}$

$$\forall v \in \mathcal{V} \quad \frac{p(v \mid y_{[i-1]})^\gamma}{\sum_{v \in \mathcal{V}} p(v \mid y_{[i-1]})^\gamma} = p(v \mid y_{[i-1]}).$$

But this would imply for every $v_s, v_r \in \mathcal{V}$ and for every $y_{[i-1]} \in \mathcal{Y}_{[i-1]}$ we have

$$\frac{p(v_s \mid y_{[i-1]})}{p(v_r \mid y_{[i-1]})} = \frac{\frac{p(v_s|y_{[i-1]})^\gamma}{\sum_{v_j \in \mathcal{V}} p(v_j|y_{[i-1]})^\gamma}}{\frac{p(v_r|y_{[i-1]})^\gamma}{\sum_{v_j \in \mathcal{V}} p(v_j|y_{[i-1]})^\gamma}} = \left( \frac{p(v_s \mid y_{[i-1]})}{p(v_r \mid y_{[i-1]})} \right)^\gamma.$$

This only happens when $\gamma = 1$ or when $\frac{p(v_s|y_{[i-1]})}{p(v_r|y_{[i-1]})} \in \{0, 1, \infty\}$. The latter of which, when seeing this needs to happen for every $v_s, v_r$, and $y_{[i-1]}$, would imply the distribution is a uniform distribution or a deterministic distribution. $\square$

### B.15 Proof of Proposition 8

*Proof.* By log properties and linearity of expectation:

$$\mathbb{E}_{y \sim p} \left[ -\log \left( \prod_{i=1}^{L} \frac{p(y_i \mid y_{[i-1]})^\gamma}{\sum_{y_j \in \mathcal{V}} p(y_j \mid y_{[i-1]})^\gamma} \right) \right] = \mathbb{E}_{y \sim p} \left[ -\sum_{i=1}^{L} \log \left( \frac{p(y_i \mid y_{[i-1]})^\gamma}{\sum_{y_j \in \mathcal{V}} p(y_j \mid y_{[i-1]})^\gamma} \right) \right] =$$
$$\sum_{i=1}^{L} \mathbb{E}_{y \sim p} \left[ -\log \left( \frac{p(y_i \mid y_{[i-1]})^\gamma}{\sum_{y_j \in \mathcal{V}} p(y_j \mid y_{[i-1]})^\gamma} \right) \right] \tag{1}$$

We will only look at one of these expectations in the sum. Choose $j \in [L]$. Then:

$$\mathbb{E}_{y \sim p} \left[ -\log \left( \frac{p(y_i \mid y_{[i-1]})^\gamma}{\sum_{y_j \in \mathcal{V}} p(y_j \mid y_{[i-1]})^\gamma} \right) \right] =$$
$$\mathbb{E}_{y \sim p} \left[ -\gamma \log \left( p(y_i \mid y_{[i-1]}) \right) + \log \left( \sum_{y_j \in \mathcal{V}} p(y_j \mid y_{[i-1]})^\gamma \right) \right] \tag{$\star$}$$

Now, we have the following inequalities:

$$\log \left( \sum_{y_j \in \mathcal{V}} p(y_j \mid y_{[i-1]})^\gamma \right) \leq \log \left( |V| \max_{y_j \in \mathcal{V}} \{ p(y_j \mid y_{[i-1]})^\gamma \} \right) = \log(|V|) + \gamma \max_{y_j \in \mathcal{V}} \{ \log \left( p(y_j \mid y_{[i-1]}) \right) \}$$

$$\log \left( \sum_{y_j \in \mathcal{V}} p(y_j \mid y_{[i-1]})^\gamma \right) \geq \log \left( |V| \min_{y_j \in \mathcal{V}} \{ p(y_j \mid y_{[i-1]})^\gamma \} \right) = \log(|V|) + \gamma \min_{y_j \in \mathcal{V}} \{ \log \left( p(y_j \mid y_{[i-1]}) \right) \}$$

$$\log \left( \sum_{y_j \in \mathcal{V}} p(y_j \mid y_{[i-1]})^\gamma \right) \geq \log \left( \max_{y_j \in \mathcal{V}} \{ p(y_j \mid y_{[i-1]})^\gamma \} \right) = \gamma \max_{y_j \in \mathcal{V}} \{ \log \left( p(y_j \mid y_{[i-1]}) \right) \}$$

Using these, notice:

$$(\star) \leq \mathbb{E}_{y \sim p} \left[ -\gamma \log \left( p(y_i \mid y_{[i-1]}) \right) + \log(|V|) + \gamma \max_{y_j \in \mathcal{V}} \{ \log \left( p(y_j \mid y_{[i-1]}) \right) \} \right] = \gamma C_1 + \log(|V|) \tag{2}$$

where $C_{1,i} \in \mathbb{Z}_+$ is a constant that only depends on $p$.

For the lower bound, by just substituting the other inequalities in we get:

$$(*) \geq -\gamma C_{2,i} + \log\left(|V|\right) \tag{3}$$
$$(*) \geq \gamma C_{3,i} \tag{4}$$

where $C_{2,i}, C_{3,i} \in \mathbb{Z}_+$ are constants that only depend on $p$.

Substituting these inequalities back into (1) will give us what we wanted to show. $\qquad\square$

We assumed $\mathcal{Y} = \mathcal{V}^L$ to allow us to use the middle inequality of the three. If we do not asumme this, then it is possible $\min_{y_j \in \mathcal{V}}\{p(y_j \mid y_{[i-1]})^\gamma\} = 0$. To then use this inequality, we would need $|V|$ to be replaced with the amount of tokens with a non-zero probability, $|V_{y_{[i-1]}}|$. This would then require taking the expectation over $\log(|V_{y_{[i-1]}}|)$ to get our bounds. We could also get a matching upper bound by doing the same with the upper bound.

