# OpenReview forum: "On Next-Token Prediction in LLMs: How End Goals Determine the Consistency of Decoding Algorithms"
_TMLR — Under review for TMLR_

### Review · Reviewer_xZDp · 2026-05-19

**Summary Of Contributions:**

The paper explores next-token prediction and decoding through the lens of surrogate-loss consistency. Specifically, by assuming that there is a correct predictor, the paper observes the consistency of decoding algorithms. This is formalized as both an information-retrieval-style correctness (N-gram Hamming loss) and sample generation (cross-entropy). Decoding strategies such as greedy, T lookahead, random, and temperature-scaled are analyzed. The main conclusion is that deterministic decoders are better aligned with point prediction style losses, and stochastic decoders are better for distribution matching objectives.

**Audience:**

Yes

**Audience Explanation:**

Most practitioners choose between greedy decoding, beam search, random sampling, and temperature scaling heuristically. Understanding the impact of these decoding techniques is an important area of research.

**Broader Impact Concerns:**

I do not see major ethical concerns that would require a separate Broader Impact Statement.

**Claims And Evidence:**

No

**Claims Explanation:**

While the paper is largely intuitive and easy to follow, I do not think the current submission fully supports all its claims.
- The definition of consistency in Section 3 uses $R(D,p^\ast,p^i_{\mathrm{ntp}},\ell) \to \inf_{h:\mathcal{X}\to\mathcal{Y}} R(h,p^\ast,\ell),$. This can only be applied to point-prediction losses such as Hamming or 0-1 loss, but not for the cross-entropy loss, where the prediction object is a distribution $p_D()$. For cross-entropy, the comparator should be over distribution-valued predictors $q()$, not deterministic $h: X \to Y$. Otherwise, this would make deterministic hypotheses have infinite cross-entropy.
- Theorem 4.2 is not fully substantiated. The proof sketch seems to consider a worst-case lower bound. This may be a valid result, but it still needs to be stated and proved as a worst-case lower bound over a class of distributions and not just as the lower bound for being optimal on the uniform distribution itself.
- Assumption 3.1 needs clarification. It states that individual conditional probabilities converge in the KL divergence. But since KL is defined between distributions, how is this translated into individual probabilities?
- Information retrieval is typically closer to exact match and calibrated decision making, not surface N-gram overlap. So, using the N-gram Hamming needs more clarification.

**Requested Changes:**

- The consistency definition for distribution-values losses must be fixed in section 3. Further, Section 5 should be rewritten using this fix.
- Theorem 4.2 and its corollary need to be worded correctly to ensure no claims are overstated.
- The KL divergence of individual probabilities needs to be clarified.
- Explaining why N-gram Hamming would fully capture information retrieval would be good.
- Optionally including other decoding methods, such as top-p beam search, would strengthen the paper.

---

> ### Author Response · Authors · 2026-05-27
>
> Thank you for your comments!
>
> We appreciate you pointing out the flaw in the way we wrote the definition of consistency. This is a flaw in our notation as, for point prediction losses, the results we prove are for any algorithm that can output sequences according to a probability distribution, which allows for randomized outputs or deterministic outputs. For cross entropy, we do implicitly use the probability distribution of the algorithm's outputs as well. We use the word "deterministic" to mean decoders that put all mass on 1 sequence output (such as greedy decoding). Thus, in Section 5, deterministic decoders would have infinite cross entropy.
>
> For Theorem 4.2, we can see how the way it is currently stated can cause confusion. We will be sure to rewrite the theorem and the proof to be more clear. What we prove is that for any algorithm A, so long as it has queried $m < |V|^L-1$ values from the net-token predictor, there exists at least two next-token probability distributions, $p_1$ and $p_2$, on which the m points queried have the same probabilities, but have different optimal sequences. Thus, our algorithm can only be optimal for one of them and thus not the other.
>
> The uniform distribution is used to show what probabilities the algorithm sees when it queries its m points, no matter what m points it chooses. Then, once it has seen these points, one can create two distributions that have the same probabilities on those m points, but have different values on the non-seen points to make them have different optimal values. Thus, no matter which output the algorithm gives, the adversarial opponent can choose the other distribution as the "true" distribution and the algorithm will not be optimal. From this, we hope the claim that no polynomial-time decoder is optimal for all probability distributions better substantiated.
>
> The definition of Assumption 3.1 is a typo as well. It should be that the next-token prediction probability distributions $p(\cdot| y_{[i-1]})$ converge to $p^*(\cdot| y_{[i-1]})$. We appreciate you pointing this out!
>
> The N-gram Hamming loss is used as a mathematically tractable proxy for some well-used metrics for evaluating correctness. As far as we are aware, N-gram metrics such as the BLEU score and ROUGE-N scores have been and are used to evaluate correctness of an output. These are, however, mathematically intractable to analyze. The N-gram Hamming loss is thus used to try keep faithful to the N-gram metrics like BLEU and ROUGE-N while having a loss function that one can work with. We note that exact match is a special case of the N-gram Hamming loss when N is the length of the output sequence.
>
> While top-p/k decoding are not explicitly discussed in the paper, they are implicitly. Since they are stochastic decoders, we know that they will not be optimal for N-gram Hamming, and since they are different than random sampling, they will not be consistent for cross entropy either. We did want to discuss them more in detail, however, due to the nature of their thresholding, to make any more non-trivial statements we believe we would need to make structural assumptions on the true next-token probabilities (e.g., if $p^\*(v_1|y_{[i-1]}) > p^\*(v_2|y_{[i-1]})$, does that imply choosing $v_1$ is always a better choice?), which is something we wanted to avoid in this paper. This same idea is also why we do not necessarily discuss beam searches either. We do agree these are interesting topics and would be interesting follow-on work.
>
> We hope these address your concerns, and if not, we look forward to the ongoing dialogue!

---

### Review · Reviewer_EMDC · 2026-05-19

**Summary Of Contributions:**

The paper's main contribution can be summarized in two parts:

1. It tries to formulate the decoding problem of LLMs into a surrogate loss consistency framework. The authors treat next-token cross entropy as the surrogate objective during training, and then analyze whether different decoding algorithms can be consistent with different sequence-level target losses. In short, it provides a theoretical view of the gap between next-token prediction and the final generation objective.

2. Building on this foundation, the authors show a dichotomy between two types of end goals. For information retrieval or correctness-oriented tasks, which are modeled by N-gram Hamming loss, deterministic decoding is generally more suitable and stochastic decoding is not consistent. For creative generation or distribution-matching tasks, which are modeled by sequence-level cross entropy, random sampling is consistent while deterministic decoding is not.

In my opinion, the positive side of this paper is that it gives a relatively clean theoretical explanation for a phenomenon that people already often observe in practice. The conclusion itself may not be surprising, but putting it into a consistency framework still makes the discussion more rigorous. I also think the negative results are somehow interesting, especially the result that even with a perfect next-token predictor, finding the globally optimal sequence under N-gram Hamming loss can still be computationally hard. This makes the paper not only a simple restatement of common decoding heuristics.

In my opinion, the core issue of this paper is that it sits in an awkward position between theory and practice. On the theory side, the framework can indeed give a more clear view of why different decoding algorithms fit different goals, but the main conclusion does not go much beyond the existing common understanding in LLM inference. On the practice side, although the idea of connecting decoding choices with end goals is reasonable, the empirical validation is far from sufficient to support its practical usefulness.

One theoretical point that I think would benefit from further clarification is the exponential lower bound in Theorem 4.2. My understanding is that the authors want to show a worst-case oracle lower bound: if a decoder wants to be optimal for all possible sequence distributions under N-gram Hamming loss, then it may need to query exponentially many next-token probabilities. This is a reasonable and potentially interesting point. However, I find the current statement a bit confusing because the theorem and proof are based on the uniform distribution. Under the exactly uniform distribution, all sequences have the same probability, so an arbitrary fixed output would already be optimal without querying exponentially many probabilities. Therefore, the hard instance seems not to be the uniform distribution itself, but rather a family of possible distributions that are indistinguishable from the queried probabilities and may contain an unqueried optimal sequence. I think it would be helpful if the authors could reformulate this result more explicitly as a worst-case oracle or decision-tree lower bound, and clearly specify the distribution family and the adversarial construction. This would make the theoretical claim much easier to verify and would also strengthen the paper’s main negative result.

The empirical part seems quite weak for me. The experiments are only based on small Markov chain simulations, where the authors compare lookahead decoding with the brute-force optimal sequence under N-gram Hamming loss. This can illustrate the theoretical claims, but it is still far from real LLM inference scenarios. Modern benchmarks such as factual QA, reasoning, code generation, or open-ended instruction following are not considered by the authors in the evaluation.

The writing is generally understandable, but I think the motivation and positioning could be made clearer. A number of claims are presented as if they have strong practical implications, but in fact many of them are already standard intuitions in decoding practice. The paper would be stronger if it more explicitly states that its main contribution is a formalization of existing intuitions, rather than suggesting that it provides direct new guidance for practical decoding.

I think the paper has the potential to be a useful theoretical contribution. However, this would require either strengthening the theoretical novelty or making the practical part much more convincing.

**Audience:**

Yes

**Audience Explanation:**

I think some researchers in the TMLR audience would be interested in this paper, especially those working on LLM inference, decoding algorithms, and theoretical understanding of autoregressive generation. The paper studies a relevant question: how the final goal of generation should affect the choice of decoding algorithm. Even though the main message is somewhat intuitive, the attempt to formalize it through consistency analysis is still useful.

**Broader Impact Concerns:**

I have not found any discussions about the limitations and potential negative societal impact. But in my opinion, this may not be a problem, since the work only focuses on theorectical understanding the decoding strategies in LLM. Still, it is highly encouraged to add corresponding discussions.

**Claims And Evidence:**

Yes

**Claims Explanation:**

Overall, I think the main claims are supported by the theoretical analysis. The paper gives a clear consistency framework and provides formal arguments for the difference between deterministic and stochastic decoding under different target losses. The Markov chain simulations also help illustrate the theoretical findings, although they are relatively simple.

That being said, I think some parts could still be made clearer. In particular, the exponential lower bound in Theorem 4.2 would benefit from a more explicit formulation. Since the proof is based on the uniform distribution, where an arbitrary sequence can already be optimal, I think the result would be easier to understand if it is stated more clearly as a worst-case oracle lower bound over a family of possible distributions. Also, the empirical evidence is mainly illustrative and does not fully show practical applicability to real LLM inference. So my answer is yes, but with some reservations about clarity and empirical strength.

**Requested Changes:**

See summary.

---

> ### Author Response · Authors · 2026-05-27
>
> Thank you for your comments!
>
> We agree that this paper's main contribution is a theoretical justification of phenomena seen in the empirical study of LLMs. We do appreciate telling us that it did not come out this way and will rewrite portions to make this more clear. Particularly, the second paragraph of the conclusion does seem to imply that our practical implications were novel. This was unintended and we will fix this.
>
> Theorem 4.2 is how you describe. We use the uniform distribution to give the values of the $m$ points the algorithm queries from the next-token predictor. Then, so long as $m < |V|^L-1$, there exists as least two probability distributions on sequences, $p_1$ and $p_2$, such that they have the same probabilities on the $m$ points seen by the algorithm, but have different values elsewhere that allow for their optimal values to be different. Thus, our algorithm can only be correct for one of them. We can see how the way it is currently written can be confusing and will make it more clear.
>
> We agree that the empirical section is weak if one wants to use it to get empirical insights on LLMs; the purpose of these was to illustrate our theoretical claims as you state. Since the theoretical results do align with the empirical results seen in the LLM literature, we did not believe it was necessary to create our own experiments.
>
> Please let us know if there are any unresolved or further questions and we look forward to any ongoing dialogue!

---

### Review · Reviewer_RVrk · 2026-06-04

**Summary Of Contributions:**

The paper investigates the interplay between decoding algorithms (greedy/lookahead decoding, random sampling, and temperature-scaled random sampling) and specific user objectives, categorized as either information retrieval or creative generation. The authors demonstrate a fundamental dichotomy where deterministic decoders are more consistent for accuracy-based tasks, while stochastic decoders are necessary for mimicking the underlying language distribution. By analyzing surrogate loss consistency, the paper proves that no single decoding strategy is optimal for every scenario. Ultimately, the findings suggest that selecting a decoder based on the intended end goal is essential for model performance. While there has been evidence for this empirically, this paper gives rigorous theoretical grounding to these results.

**Audience:**

Yes

**Audience Explanation:**

Absolutely, the paper deals with a very hot topic in LLM research and should be of relevance to a wide audience.

**Broader Impact Concerns:**

No broader concerns.

**Claims And Evidence:**

Yes

**Claims Explanation:**

Overall, I am positive about this work. The paper provides a much-needed theoretical framework for something the field has largely treated as an empirical "art": the choice of decoding algorithms. It successfully bridges the gap between classic surrogate loss consistency theory and modern LLM generation. It unveils an inherence dichotomy between accuracy-based and creative tasks, and argues why no decoding algorithm is perfect but the best choice largely depends on the task at hand. The paper is technical in some parts - as far as I was able to check, the proofs appear to be correct.

Some concerns:
- The simulations use very small Markov chains. I wasn't clear whether these results can accurately reflect performance sub-optimality when using large multibillion-parameter Transformer models.
- The proof for Theorem 4.2 relies entirely on black-box access to the next-token predictor. However, in Transformer architectures we also have access to internal activations, weights, and gradients. I was wondering whether polynomial decoding algorithms that make use of this white-box information may be possible. Does the current analysis extend beyond black-box models?
- Proposition 1 allows the authors to analyze decoders assuming the model has already perfectly converged. It requires the loss function to be M-bounded. This assumption is true for the N-gram Hamming loss. But given the log-loss can be infinite for deterministic decoders (as the authors acknowledge), the "boundedness" required for the proof of Proposition 1 does not technically hold for cross-entropy in the same way.
-  Analyzing consistency on distributions more complex than random Dirichlet samples (e.g., power-law distributions) would make the findings more applicable to practitioners.

**Requested Changes:**

I feel this work introduces an elegant theoretical framework to establish a number of key points. The limitations/weaknesses I previously mentioned are not critical, but it would be nice if the authors could address them in the paper or at least discuss them in their response.

 A few additional points:
- Propositions 3 and 4 show that increasing lookahead (K) or decreasing the tokens kept (T) does not monotonically improve performance. This is a very interesting observation. Natural follow-up question: why and when looking further ahead actually leads a decoder into a worse path?
- Did the authors contemplate about extending their analysis to Top-K or Nucleus (Top-p) sampling? Are there perhaps any specific challenges associated with these settings?
- The authors frequently restrict their proofs to a specific class of distributions, where there are no ties in the arg max at any step. Isn't this a limitation for practical LLMs? The strategy for tie-breaking (random vs. deterministic) can fundamentally change whether an algorithm is considered "consistent" in a practical sense.

---

> ### Author Response · Authors · 2026-06-07
>
> Thank you for your comments and kind words!
>
> For the Markov chains experiment, they were mainly to illustrate our theory with a simple experiment. While our theoretical findings do support what has been seen empirically in the literature, the results of this particular experiment themselves may or may not be representative of actual LLMs due to its simplicity. Running this experiment on an actual LLM would be difficult due to the size of the vocabularies of LLMs (a small LLM with a vocabulary size of ~5000 means there are $ 3.1\times 10^{18} $ different 5 word sentences we would need to check for optimality). Thus, since this is mainly a theory paper and the findings support what has been seen in the current literature, we figured a simple experiment would suffice for the purposes of this paper.
>
> If given white-box access, there can be cases where Theorem 4.2 would break if one is able to infer extra information on the values of $p(v_i |y_{[i-1]})$ from other $p(v|y)$ values. For example, if we have a model where the next token probability only depends on where it is in the sequence:
> $$ \forall i \in [L-1], \forall y_1,y_2 \in \mathcal{Y},  \text{length}(y_1) = \text{length}(y_2) = i, \forall v \in \mathcal{V} \quad p(v|y_1) = p(v|y_2), $$
> then we only need to query $|\mathcal{V}-1| $ for each spot in the sequence, giving us only $O(L|\mathcal{V}-1|)$ runtime for finding optimality. However, for sufficiently complex models where simplifications such as above  can not happen, this would be equivalent to our black-box set up.
>
> We completely agree that the cross entropy sequence loss function would violate the assumptions for Proposition 1 and thus we were careful to not use Proposition 1 in any of the proofs that use the cross entropy loss function on the sequence output.
>
> For the followup question on Propositions 3 and 4: we believe this is due to them being greedy algorithms. Since any greedy algorithm can only look so far ahead, they all just have to make their best guess with what they have seen so far. Thus, once these guesses diverge from each other, one has to be wrong and since they each do not know all information, one can create adversarial distributions to make either one correct. A real-life example can be trying to drive from one place to another. If the place you are trying to drive to is just off a highway, but the roads to get to the highway are backed up, you might be inclined to take a secondary route to get to your destination. However, if you know that all but the last mile of the highway itself is clear, you would be more inclined to take the highway route. However, you did not know there is standstill traffic at the last mile and it would have been quicker to take the secondary route.
>
> While we do not explicitly discuss top-p/k sampling in the paper, we do implicitly discuss them. Since they are stochastic decoders, we know for the N-gram Hamming loss they will not be optimal. Also, since they are not random sampling, they will not be optimal for the cross entropy loss as well. We would like to have done more with them as they are well-used decoders, however, we felt as if any more non-trivial statements would require assumptions on our true data generating distributions on the usefulness of removing low-probability tokens, which we did not want to add. We do agree that these are worthwhile to study and would be interesting follow-up material.
>
> For the question on ties, we hope in practice that there is enough precision in the probabilities that ties happen very rarely. We do agree that if they do happen, then any way of tie-breaking will dramatically change outputs, which is a problem if one is trying to be correct. If a greedy decoder algorithm (such as $K_T$-lookahead) breaks ties deterministically, then there are adversarial true probability distributions that will make it choose incorrectly. If it breaks ties randomly, then it can not be consistent since there is a non-zero chance it will choose the wrong output. Thus, if ties do happen, it is basically a lose-lose situation for the decoder in terms of being optimal for an unknown true probability distribution.
>
> Please let us know if you have any more thoughts or questions and we look forward to any further dialogue!